# RAD54 promotes alternative lengthening of telomeres by mediating branch migration

Emily Mason-Osann, Katherine Terranova, Nicholas Lupo, Ying Jie Lock, Lisa M Carson & Rachel Litman Flynn[*] (iD)

## Abstract

**Cancer cells can activate the alternative lengthening of telomeres (ALT) pathway to promote replicative immortality. The ALT pathway promotes telomere elongation through a homologous recombination pathway known as break-induced replication (BIR), which is often engaged to repair single-ended double-stranded breaks (DSBs). Single-ended DSBs are resected to promote strand invasion and facilitate the formation of a local displacement loop (D-loop), which can trigger DNA synthesis, and ultimately promote telomere elongation. However, the exact proteins involved in the maturation, migration, and resolution of D-loops at ALT telomeres are unclear. *In vitro*, the DNA translocase RAD54 both binds D-loops and promotes branch migration suggesting that RAD54 may function to promote ALT activity. Here, we demonstrate that RAD54 is enriched at ALT telomeres and promotes telomeric DNA synthesis through its ATPase-dependent branch migration activity. Loss of RAD54 leads to the formation of unresolved recombination intermediates at telomeres that form ultra-fine anaphase bridges in mitosis. These data demonstrate an important role for RAD54 in promoting ALT-mediated telomere synthesis.**

**Keywords** alternative lengthening of telomeres; break-induced replication; RAD54; telomeres

**Subject Categories** DNA Replication, Recombination & Repair

## Introduction

Cancer cells must maintain telomere length in order to escape crisis and overcome cellular senescence. The majority of cancer cells do so through reactivation of telomerase however, ~ 5–10% of cancers maintain telomere length in the absence of telomerase using the alternative lengthening of telomere (ALT) pathway [1–4]. ALT is a recombination-based pathway whereby one telomere uses either another telomere or extrachromosomal telomeric repeats as a template for elongation via break-induced replication (BIR) [2,3,5,6]. BIR is the preferred pathway for repairing single-ended

DNA double-stranded breaks (DSBs), a structure often formed following the collapse of irreversibly stalled replication forks throughout the genome. ALT cells undergo chronic replication stress and, consequently, exhibit high levels of spontaneous telomeric DNA damage [7–10]. In addition to persistent DNA damage, ALT-positive human cancer cells are defined by several unique cellular phenotypes including heterogeneous telomere length, nuclear phase-separated condensates called ALT-associated PML bodies (APBs), and C-rich extrachromosomal telomeric repeats (C-circles) [2,3,11–13].

While telomere elongation via ALT has long been known to rely on homology-directed repair, it was recently demonstrated to specifically resemble BIR in mammalian cells. BIR at ALT telomeres can proceed via both RAD51-dependent and RAD51-independent mechanisms to facilitate telomere elongation [5,6,14]. RAD51 functions to promote the search and capture of homologous template DNA during recombination and may also function to stabilize replication forks during replication stress [15]. However, RAD51 is not essential for BIR at ALT telomeres as RAD52 may also function to promote the homologous pairing of DNA during ALT telomere elongation [5,6,16]. Whether RAD51- or RAD52-dependent, the process of search and capture facilitates the formation of a jointed DNA structure called a displacement loop (D-loop), which functions as the platform for recruitment of DNA polymerases and the initiation of DNA synthesis. Synthesis events at ALT telomeres are thought to rely on the sequential recruitment of the DNA polymerases Polη and Polδ [17]. Polη has previously been shown to extend D-loops *in vitro* [18], providing evidence that Polη may function to initiate, or prime, DNA synthesis. Following the initiation of DNA synthesis, the POLD3 subunit of Polδ replaces Polη and synthesizes longer tracts of DNA to promote telomere extension events [6,17]. These extended D-loops must then be processed to resolve the joint molecules and ensure completion of telomere elongation.

Recombination intermediates formed at ALT telomeres must be processed prior to mitosis to maintain chromosomal stability [19]. The resolution of recombination intermediates at ALT telomeres proceeds either through nucleolytic cleavage by resolving enzymes such as the SMX complex (SLX1-4, MUS81-EME1, XPF-ERCC1) or through the migration of branched DNA structures leading to dissolution by the dissolvasome complex BTR (BLM, TOP3α, RMI1, RMI2) [20]. It was recently described that SLX4IP binds to both

Departments of Pharmacology & Experimental Therapeutics, Medicine Cancer Center, Boston University School of Medicine, Boston, MA, USA
*Corresponding author. Tel: +1 617 358 4666; E-mail: rlflynn@bu.edu

BLM and SLX4 to help regulate the balance between dissolution and cleavage at ALT telomeres [21]. Resolution by nucleolytic cleavage generates cross-over events in the absence of net telomere elongation. Conversely, branch migration promotes DNA synthesis that, when followed by dissolution and/or cleavage, generates telomere extension events. Therefore, defining the enzymes involved specifically in branch migration-dependent dissolution of D-loops at ALT telomeres will provide insight toward the regulation of elongation events at ALT telomeres and potentially identify novel therapeutic targets to inhibit telomere length maintenance in ALT-positive cancers.

RAD54 is a member of the RAD52 epistasis group and belongs to the SWI2/SNF2 family of ATPases [22,23]. Early studies demonstrated that RAD54 is a DNA-dependent ATPase that translocates along dsDNA, suggesting a role for RAD54 in homologous recombination. Generally, homologous recombination can be broken up into three main stages, the processing of DNA for the homology search (pre-synapsis), the formation of a mature D-loop (synapsis), and the dissolution or resolution of joint molecules (post-synapsis). Since the initial identification of RAD54, the biochemical activities of RAD54 have been studied extensively *in vitro* and demonstrate a range of functions from pre-synapsis to post-synapsis during homologous recombination [22,23].

During pre-synapsis, a broken DNA end is resected to form a 3′ single-stranded overhang. This overhang is then bound by RAD51 to create a nucleoprotein filament that promotes strand invasion during the search for a homologous template. Previous studies have demonstrated that RAD54 interacts with RAD51 to stabilize the RAD51 nucleoprotein filament and to stimulate both strand invasion and the formation of the D-loop during synapsis [24,25]. The ability of RAD54 to stimulate strand invasion relies on its ATPase activity, suggesting that RAD54 may function to regulate the accessibility of the template DNA, either by inducing topological changes (i.e., supercoiling) or by facilitating nucleosome repositioning [26]. Once a homologous template has been found, RAD54 has been shown to disrupt the RAD51 nucleoprotein filament, promoting the removal of RAD51 and the subsequent conversion of a paranemic DNA joint into a fully synapsed plectonemic joint [27–29]. Thus, *in vitro*, RAD54 functions to promote the formation, and/or synapsis, of a mature D-loop that is accessible to DNA polymerases and capable of being extended during homologous recombination [30–32].

In addition to the proposed roles for RAD54 during synapsis, data also suggest a role for RAD54 in post-synaptic processing of D-loops. RAD54 binds to branched recombination intermediates and promotes their branch migration in an ATPase-dependent manner [33,34]. While RAD54 can bind and branch migrate many different DNA structures, it has highest affinity for partial-X junctions, the specific structure formed during BIR [34,35]. The branch migration activity of RAD54 can dissolve recombination intermediate structures. Unlike other branch migrating enzymes, such as BLM and RECQL, RAD54 has been shown to be uniquely efficient at bypassing sites of DNA heterology. For example, RAD54 can dissolve a D-loop that has formed with mismatches between the invading strand and template DNA [36]. Together, these findings suggest that RAD54 not only regulates the formation and resolution of recombination intermediates, but that RAD54 may also ensure the fidelity of homologous recombination by regulating heterologous strand invasion events.

Previous studies have demonstrated that RAD54 functions to regulate telomere length maintenance in murine cells [37,38]; however, the contribution of RAD54 specifically to ALT-mediated telomere elongation in human cancer has not yet been investigated. Given the role of RAD54 in regulating the formation and resolution of recombination intermediates, we asked whether RAD54 might also function at ALT telomeres to regulate BIR. Here, we demonstrate that RAD54 is recruited to telomeric DNA in response to DNA damage and functions to promote ALT activity via BIR. Depletion of RAD54 led to a decrease in DNA synthesis at ALT telomeres. The ability of RAD54 to promote DNA synthesis at ALT telomeres was dependent on both its ATPase and branch migration activities. Moreover, combined depletion of RAD54 and the resolvase enzyme SLX4 led to the formation of unresolved recombination intermediates visualized as ultra-fine anaphase bridges in mitosis. Taken together, our data highlight a post-synaptic function for RAD54 during the dissolution of recombination intermediates at ALT telomeres.

# Results and Discussion

### RAD54 localizes to ALT telomeres and responds to telomeric DNA damage

Given the crucial role of RAD54 in homology-directed repair processes in eukaryotic cells, we asked whether RAD54 functions at ALT telomeres to promote break-induced replication. To address this, we first determined whether RAD54 associated with telomeres in a panel of unperturbed human cancer cell lines using combined immunofluorescence (IF) and DNA fluorescence *in situ* hybridization (FISH). Here, using IF-FISH we demonstrate that RAD54 colocalized with telomeric DNA across a panel of ALT-positive osteosarcoma cell lines. Moreover, the colocalization between RAD54 and telomeric DNA was enriched in ALT-positive cells as compared to the colocalization events in telomerase-positive cells (Fig 1A and B). In ALT cells, telomeres are heterogeneous in length, including very long telomeres that can exacerbate replication stress [2]. The observed enrichment of RAD54 at ALT telomeres was not simply a consequence of the extended length of ALT telomeres as we were unable to detect RAD54 at telomeric DNA in the HeLa 1.2.11 (HeLa LT) cell line that maintains long telomeres (Fig 1A and B). Given that ALT telomeres are frequently associated with DNA repair factors in specific ALT-associated PML bodies (APBs) [11], we asked whether the accumulation of RAD54 at ALT telomeres was specific to APBs. In fact, we found that the majority of RAD54 foci detected by IF in ALT cells colocalized with telomeres in APBs (Fig 1C and D), suggesting that RAD54 may be contributing to the ALT mechanism.

ALT telomeres are associated with replication stress and spontaneous DNA damage. Therefore, we asked whether RAD54 was recruited to ALT telomeres in response to DNA damage [5,6]. To test this, we first treated ALT cells with the topoisomerase inhibitor camptothecin (CPT) to induce the formation of DNA breaks. Treatment with CPT led to an increase in RAD54 foci formation and a significant enrichment of RAD54 at ALT telomeres (Fig EV1A and B). Given that CPT induces DNA damage throughout the genome, we asked whether we could further enhance RAD54 recruitment to ALT telomeres by inducing DNA damage specifically at the telomeres. Here, we generated DSBs at telomeric DNA by inducing

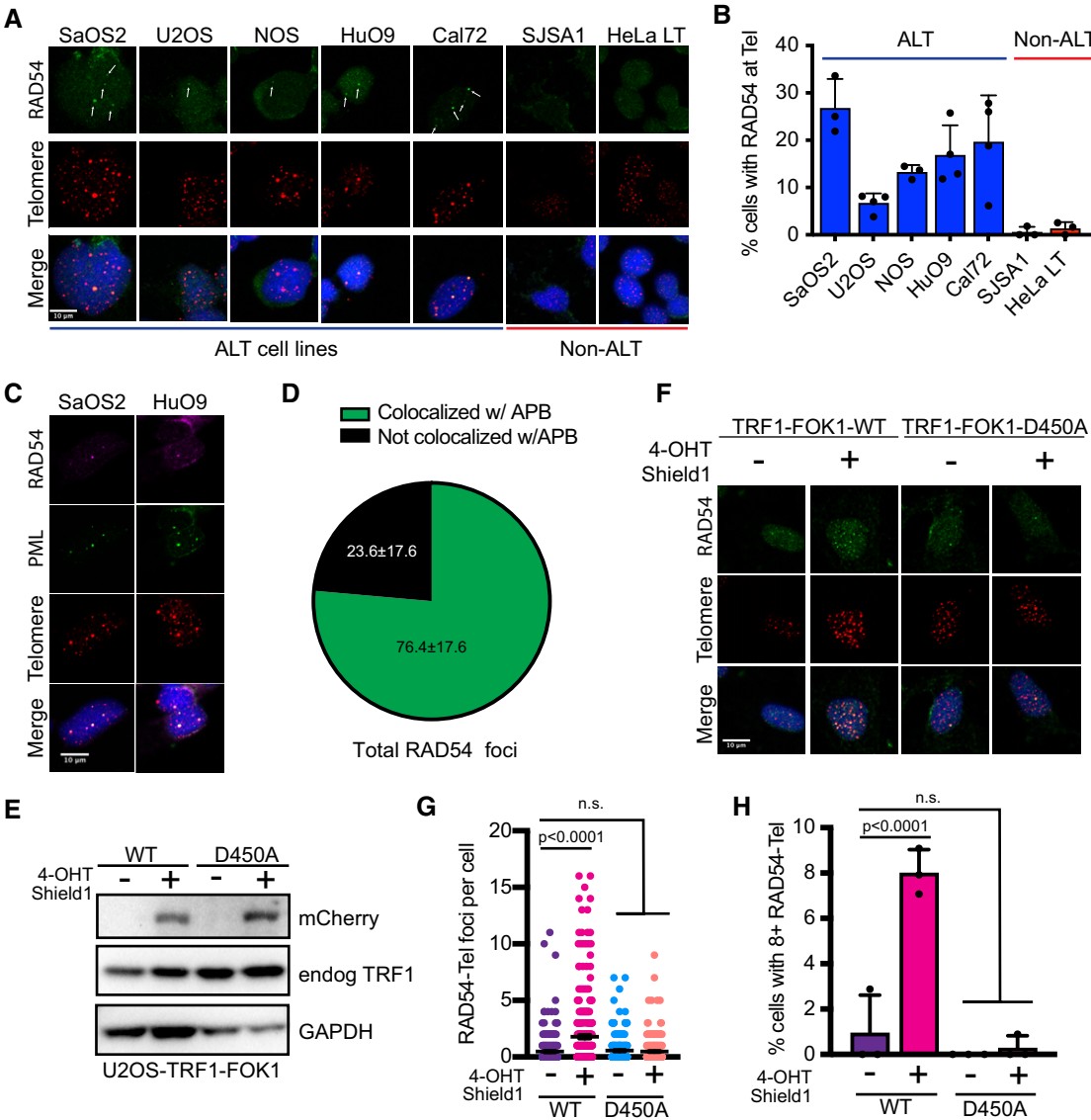

**Figure 1. RAD54 localizes to ALT telomeres in response to DNA damage.**

A Combined IF and DNA FISH analysis of RAD54 (IF) and telomeres (FISH) in ALT and non-ALT cell lines. White arrows indicate RAD54 foci that colocalize with telomeres. Scale bars = 10 μm.

B Quantification of data in A. A cell was counted positive if it contained 1 or more colocalization event between RAD54 and the telomere. At least 100 cells were counted per cell line per repeat. For SaOS2, NOS, SJSA1, HeLa LT *n* = 3. For Cal72, U2OS, HuO9 *n* = 4. Values shown are mean ± SD.

C Combined IF and DNA FISH analysis of RAD54 (IF), PML (IF), and telomeres (FISH) in ALT cell lines SaOS2 and HuO9. Scale bars = 10 μm.

D Quantification of HuO9 data from C. RAD54 foci were detected using particle analysis in Fiji. Total RAD54 foci were counted. RAD54 foci were considered in APBs if they colocalized with both PML and telomere. At least 100 cells counted per repeat, *n* = 3. Values shown are mean ± SD.

E Western blot showing expression of TRF1-FOK1-mCherry fusion protein following doxycycline induction, both WT and D450A (nuclease dead). Cells were treated with 40 ng/ml doxycycline for 16 h and then 1 μM Shield-1 and 1 μM 4-OHT for 4 h.

F Combined IF and DNA FISH for RAD54 (IF) and telomeres (FISH) in U2OS-TRF1-FOK1-WT or U2OS-TRF1-FOK1-D450A (nuclease dead) cells. Cells were treated as in E immediately prior to staining. Scale bars = 10 μm.

G Quantification of data from F. RAD54 foci were detected using particle analysis in Fiji. At least a total of 300 cells were contained from 3 repeated experiments. Values shown are mean ± SEM. Values were compared using Kruskal–Wallis test followed by Dunn's multiple comparison test.

H Quantification of data from F. RAD54 foci were detected using particle analysis in Fiji. Cells were counted as positive if they counted 8 or more RAD54 foci that colocalized with telomeres. Values shown are mean ± SEM of *n* = 3 biological replicates. Values were compared using standard two-way ANOVA followed by Sidak's multiple comparison test.

expression of the chimeric TRF1-FOK1 protein [6,14]. Following incubation with doxycycline, tamoxifen, and Shield1, we could detect the expression of TRF1-FOK1-mCherry protein by Western blot (Fig 1E). Moreover, expression of TRF1-FOK1-mCherry led to an obvious increase in telomeric foci size measured by FISH, indicative of the formation of telomeric DNA DSBs and subsequent

telomere clustering events [14]. Following expression of TRF1-FOK1-mCherry, we also observed a significant increase in the number of RAD54 foci at telomeric DNA and a significant increase in the percentage of cells containing 8 or more RAD54 foci that colocalized to telomeric DNA (Fig 1F–H). In contrast, expression of TRF1 fused to the nuclease-dead FOK1 mutant protein (D450A) caused no change in the localization of RAD54 to telomeres (Fig 1F–H). Together, these data demonstrate that RAD54 is localized at ALT telomeres, and this localization is enriched in response to DNA damage, suggesting that RAD54 may function to regulate the ALT pathway.

### RAD54 is dispensable for synapsis at ALT telomeres

Previous studies suggest both a pre-synaptic and a post-synaptic role for RAD54 in the regulation of homologous recombination [33–35,39]. Pre-synaptically, RAD54 has been demonstrated to regulate RAD51 nucleoprotein filament formation and stability [22,24,29,40–42]. Given that BIR at ALT telomeres has been demonstrated to promote telomere elongation via both RAD51-dependent and RAD51-independent mechanisms [6,14], we asked whether RAD54 functioned to regulate RAD51 at ALT telomeres. Depletion of RAD54 by siRNA in ALT cell lines did not lead to statistically significant changes in the localization of RAD51 to ALT telomeres as measured by combined IF-FISH (Figs 2A and B, and EV2A–C). Likewise, loss of RAD54 did not affect the localization of RAD51 to telomeric DNA by chromatin immunoprecipitation (ChIP) (Figs 2C–E and EV2D). Given the known interaction between RAD51 and RAD54 at sites of RAD51-mediated synapsis [29], we also wanted to determine whether RAD51 was involved in recruiting RAD54 to possible sites of D-loop formation at ALT telomeres. We found that depletion of RAD51 did not affect RAD54 localization to ALT telomeres (Fig EV2E–G) suggesting that RAD51 is not essential for RAD54 function at ALT telomeres.

DNA strand invasion and subsequent synapsis promote the formation of a plectonemic DNA molecule. This plectonemic structure is required to ensure polymerase binding, facilitate DNA synthesis at the displacement loop (D-loop), and ensure elongation of the invading strand [31]. PCNA recruitment to ALT telomeres is an early event in the ALT mechanism, recruiting DNA polymerases and establishing a platform for post-synaptic DNA synthesis events at ALT telomeres [6]. Expanding on previous data [6], we demonstrate that spontaneous PCNA localization to the telomeres is found across a panel of ALT-positive cells (Fig EV2H and I). Depletion of RAD54 did not prevent PCNA recruitment to ALT telomeres (Fig EV2J and K), but in fact caused a significant increase in PCNA recruitment to telomeres in at least one ALT-positive cell line suggesting that RAD54 is not essential for the formation of the mature plectonemic structure. Several other proteins have been reported to not only regulate synapsis and/or D-loop formation but also regulate ALT activity including, RAD51AP1, PALB2, and HOP2-MND1, suggesting that additional recombination factors may compensate for RAD54 loss at ALT telomeres [6,17,43].

PCNA recruitment to ALT telomeres has been proposed to facilitate BIR-mediated telomere elongation by recruiting the DNA polymerases Polη and Polδ. Polη is thought to prime DNA synthesis during ALT telomere elongation by generating short tracts of DNA [17,18], whereas Polδ functions to ensure the synthesis of long tracts of DNA at ALT telomeres [6,17]. RAD54 has been demonstrated to regulate the formation of the mature plectonemic D-loop and, ultimately, promote DNA synthesis at the 3′ end of the invading strand [29,31]. Although we did not detect changes in PCNA localization at ALT telomeres, we were curious whether loss of RAD54 led to defects in either the recruitment of, and/or priming by, Polη. Given that Polη can only bind mature D-loops, we analyzed the recruitment of Polη-GFP to telomeric DNA by IF-FISH as a surrogate for D-loop formation. Overexpression of Polη can be toxic in cells;[17] therefore, we only scored viable cells positive for GFP expression. Here, we demonstrate that loss of RAD54 did not lead to a significant change in the percentage of GFP-positive cells that contained colocalization of Polη-GFP foci with ALT telomeres (Fig 2F and G). These data suggest that RAD54 is not required for the formation of the mature plectonemic joint nor for recruitment of Polη to ALT telomeres during BIR-mediated telomere elongation.

### RAD54 promotes ALT activity

Polη priming is followed by long-range DNA synthesis and productive elongation by POLD3 [6,17]. While the recruitment of Polη was unaffected by loss of RAD54, we were curious whether long-range synthesis remained intact. Therefore, we analyzed EdU incorporation at ALT telomeres using Click-it chemistry combined with telomere FISH to visualize nascent synthesis events at the telomeres

**Figure 2. RAD54 is dispensable for synapsis at ALT telomeres.**

A   Combined IF and DNA FISH for RAD51 (IF) and telomeres (FISH) in SaOS2 or HuO9 cells. Cells were transfected with 20 nM siRAD54#2 for 48 h immediately prior to staining. Scale bars = 10 μm.

B   Quantification of A. A cell was counted as positive if it contained at least 1 colocalization event between RAD51 and telomeres. At least 100 cells were counted per condition per repeat, *n* = 3. Values shown are mean ± SD. Data were compared using standard two-way ANOVA followed by Sidak's multiple comparison test.

C   ChIP for telomeric DNA associated with RAD51. DNA dot blot probed with DIG-labeled telomere probe or Alu probe. HuO9 cells were transfected with 20 nM siRAD54 #2 or 15 μg myc-TRF2 for 72 h prior to immunoprecipitation with RAD51 antibody (Mock, siRAD54 conditions) or myc antibody (myc-TRF2 positive control).

D   Quantification of (CCCTAA)$_4$ dot blot performed in C. IP values were normalized to loading (IP/input) and then normalized to mock control. Values shown are mean ± SD of *n* = 3 biological replicates. Values were compared using two-tailed unpaired Student's *t*-test.

E   Western blot of samples used for ChIP in C, D.

F   Combined IF and DNA FISH for GFP (IF) and telomeres (FISH) in SaOS2 and HuO9 cells. Cells were forward transfected with Polη-GFP and then after 24 h transfected with 20 nM siRAD54#2. Cells were stained for GFP and telomeres 48 h after siRNA transfection. Scale bars = 10 μm.

G   Quantification of data in F. GFP-negative cells were excluded from analysis. Cells were considered positive if they showed 1 or more colocalization events between GFP (Polη) and telomeres. At least 50 GFP-positive cells were counted per condition per repeat, *n* = 3. Values shown are mean ± SD. Values were compared using standard two-way ANOVA followed by Sidak's multiple comparison test.

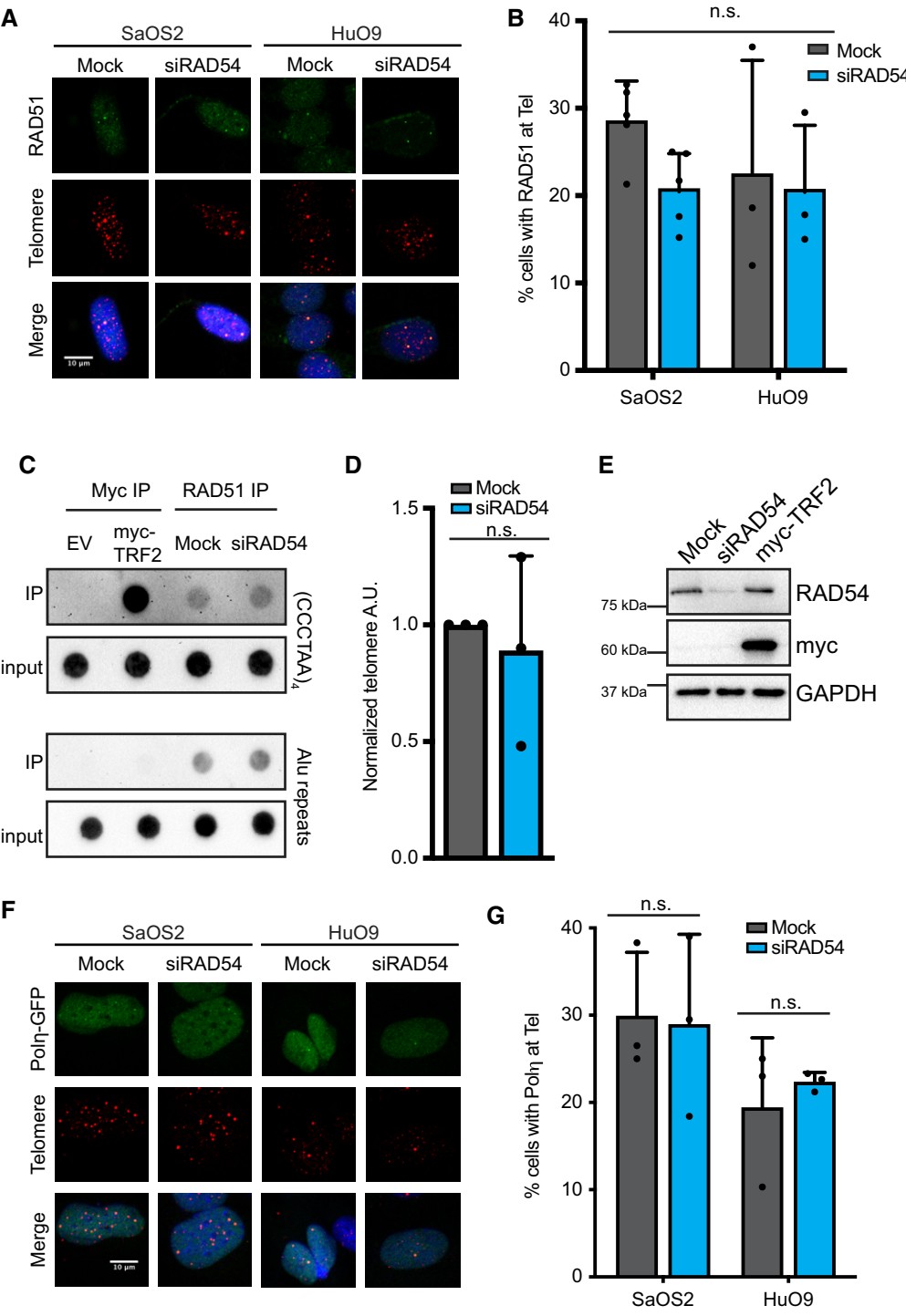

**Figure 2.**

[7,8,44]. To exclude S-phase DNA synthesis events, we only scored cells that lacked pan-nuclear EdU staining or contained fewer than 8 EdU foci. RAD54 depletion led to a significant reduction in EdU incorporation events that colocalized with the telomeric DNA without altering cell cycle progression (Figs 3A and B, and EV3A and B), indicating that RAD54 promotes nascent telomere repeat synthesis at ALT telomeres.

The defects in DNA synthesis following RAD54 depletion suggest that RAD54 functions to promote ALT activity. Therefore, we asked whether RAD54 depletion would also lead to a decrease in other hallmarks of ALT activity. RAD54 depletion by siRNA in ALT-positive cancer cells led to a significant decrease in C-rich extrachromosomal telomeric repeat DNA (C-circles) (Fig 3C and D) [45]. C-circles correlate with ALT activity and are thought to arise

as a byproduct of the recombination reaction at ALT telomeres [12,13]. To determine whether this decrease in C-circles correlated with an increase in unresolved recombination intermediates, we analyzed APB formation. APBs are induced by DNA damage and colocalize with DNA damage repair factors leading to the speculation that APBs function as platforms for the recombination of ALT telomeres [11,44]. Moreover, APBs are increased in the absence of key repair enzymes, suggesting that defects in the resolution of DNA damage may lead to the accumulation of APBs [20,21]. Likewise, we found that loss of RAD54 led to a modest yet significant

increase in APB-positive cells, supporting the idea that loss of RAD54 leads to an increase in stalled or irreparable recombination intermediates (Figs 3E and F, and EV3C).

### RAD54 branch migration promotes DNA synthesis at ALT telomeres

Branch migration of the mature D-loop, including extension and dissolution of the resulting recombination intermediate, is a post-synaptic event in BIR. RAD54 is a DNA translocase that relies on its

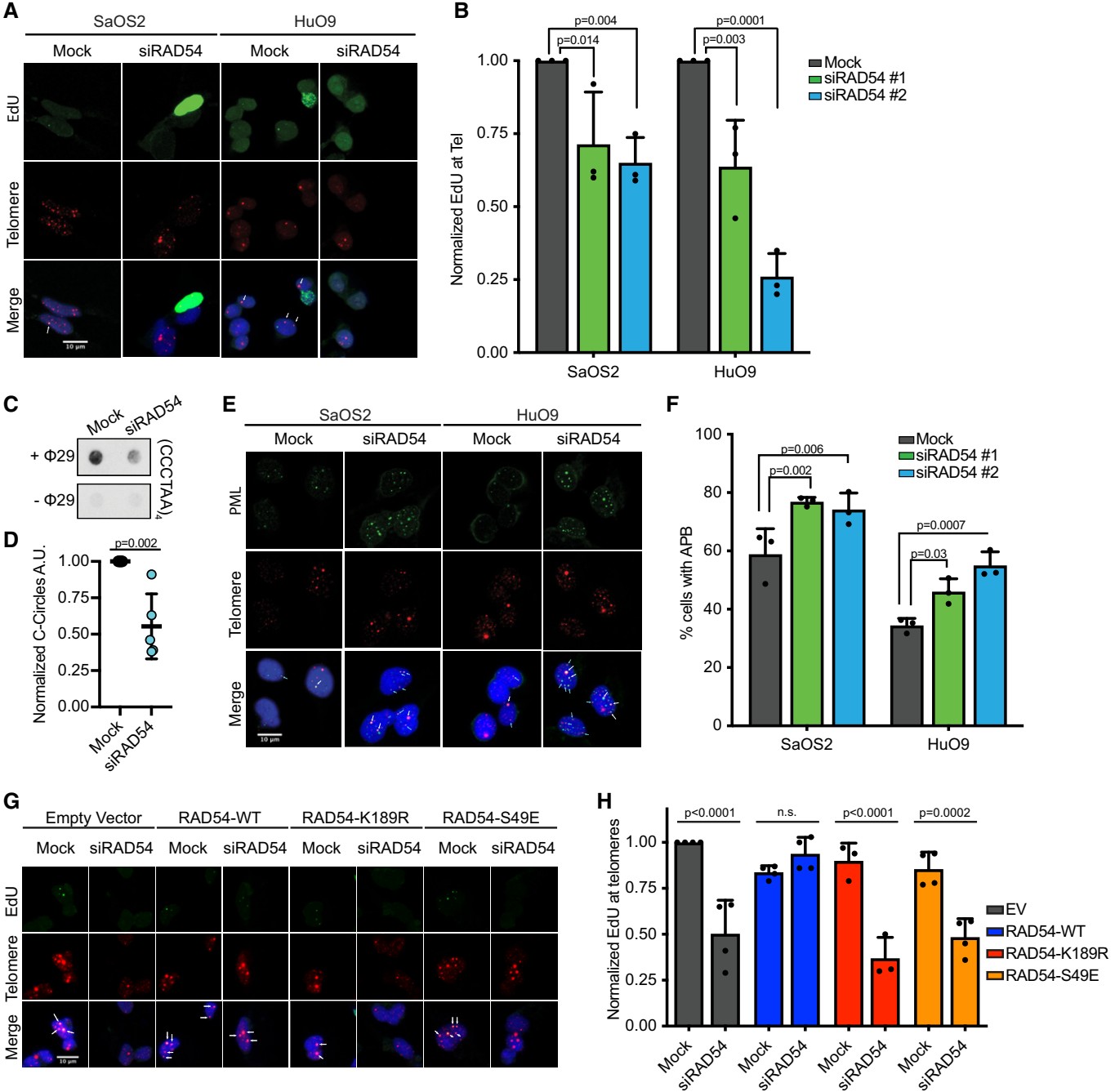

**Figure 3.**

**Figure 3.  RAD54 oligomerization and ATPase activity regulate break-induced replication at ALT telomeres.**

A   Representative images of combined EdU staining (Click-it) and DNA FISH for telomeres. HuO9 or SaOS2 cells were transfected with 20 nM siRAD54#2 (representative images for siRAD54#1 not shown) for 48 h followed by a 1.5-h pulse of EdU. Pan-nuclear EdU stain represents S-phase cells. White arrows indicate non-S-phase cells containing EdU colocalizing with telomeres. Scale bars = 10 μm.

B   Quantification of data shown in A. HuO9 or SaOS2 cells were mock-transfected or transfected with 20 nM siRAD54#1 or 20 nM siRAD54#2 for 48 h followed by a 1.5-h pulse of EdU. Cells with pan-nuclear EdU signal or greater than 8 EdU foci were excluded from analysis as S-phase cells. Non-S-phase cells were considered positive if they contained at least one EdU foci colocalizing with telomeres. Data were normalized to mock condition for each repeat. At least 100 non-S-phase cells were counted per condition per repeat, $n = 3$. Values shown are mean ± SD. Data were analyzed using two-way ANOVA followed by Dunnett's multiple comparison test.

C   Representative DNA dot blot from C-circle assay on HuO9 cells, either mock-transfected or transfected with 20 nM siRAD54#2 for 48 h.

D   Quantification of C-circle assay in C. Signal was quantified with densitometry, background (−Φ29) was subtracted, and signal was normalized to mock. $N = 5$ biological replicates values shown are mean ± SD. Conditions were analyzed using unpaired two-tailed Student's *t*-test.

E   Representative images of combined IF and DNA FISH for PML (IF) and telomeres (FISH). HuO9 or SaOS2 cells were mock-transfected or transfected with 20 nM siRAD54#2 for 48 h. White arrows indicate large colocalization events. Blue arrows indicate small colocalization events. Scale bars = 10 μm.

F   Quantification HuO9 or SaOS2 cells were mock-transfected or transfected with 20 nM siRAD54#1 or 20 nM siRAD54#2 for 48 h. Cells were considered positive if they contained 1 large or 3 small colocalization events. At least 100 cells were counted per condition per repeat, $n = 3$. Values shown are mean ± SD. Data were analyzed using two-way ANOVA followed by Dunnett's multiple comparison test.

G   Representative images of combined EdU staining (Click-it) and DNA FISH for telomeric DNA. HuO9 cells were forward transfected with 2 μg empty vector (EV), RAD54-WT, RAD54-K189R, or RAD54-S49E construct. After 24 h, cells were transfected with 20 nM siRAD54#2 for 48 h followed by a 1.5-h pulse of EdU. White arrows indicate non-S-phase cells containing EdU colocalizing with telomeres. Scale bars = 10 μm.

H   Quantification of data shown in G. Non-S-phase cells were considered positive if they contained at least one EdU foci colocalizing with telomeres. Data were normalized to EV-Mock condition for each repeat. At least 100 non-S-phase cells were counted per condition per repeat, $n = 4$ for EV, WT, S49E, $n = 3$ for K189R. Values shown are mean ± SD. Data were analyzed using two-way ANOVA followed by Sidak's multiple comparison test.

ability to both oligomerize and hydrolyze ATP to promote branch migration [33,34,41,46]. Oligomerization of RAD54 is regulated through a phosphorylation event on the N-terminal domain at Serine-49 [6]. CDK2 phosphorylation of S49 inhibits the oligomerization and subsequent branch migration of RAD54, leaving other RAD54 functions intact, such as its ability to bind to DNA and stimulate RAD51-mediated strand invasion [6]. By overexpressing an siRNA-resistant RAD54 construct (Fig EV3D and E), we were able to rescue the nascent telomere synthesis defect caused by RAD54 depletion, as measured by EdU incorporation events (Fig 3G and H). In contrast, neither the ATPase dead RAD54 mutant (K189R) [47] nor the phosphomimetic RAD54 mutant (S49E) [46] of RAD54 were able to rescue the EdU incorporation defect caused by RAD54 depletion (Fig 3G and H). These data demonstrate that RAD54 oligomerization and ATPase activity are critical for branch migration of the mature D-loop further highlighting a role for RAD54 in the post-synaptic regulation of BIR at ALT telomeres.

## RAD54 limits the formation of ultra-fine anaphase bridges at ALT telomeres

It was previously suggested that the recombination intermediates formed during BIR at ALT telomeres are processed by the BTR complex (BLM, TOP3A, RMI1/2) to promote dissolution and telomere elongation events, and/or by the SMX resolvasome (SLX1-4, MUS81-EME1, XPF-ERCC1) to promote cleavage and telomere cross-over events without elongation [17,20]. If loss of RAD54 led to the formation of unresolved recombination intermediates, we would expect to see an increase in the recruitment of resolving enzymes to these sites of recombination to process the joint molecules prior to mitosis. Therefore, we specifically analyzed the recruitment of BLM and MUS81 to telomeres following RAD54 knockdown by IF-FISH. Although RAD54 depletion did not affect BLM recruitment to ALT telomeres (Fig 4A and B), it did lead to a significant increase in MUS81 at ALT telomeres (Figs 4C and D, and EV4A–C). These findings raised the possibility that ALT cells may rely on SMX-mediated cleavage and telomere sister chromatid exchange events (T-SCE) to

resolve DNA intermediates in the absence of RAD54. Consistent with this prediction, we demonstrate a significant increase in T-SCE events following RAD54 depletion (Fig 4E and F), highlighting RAD54 function in telomere dissolution during BIR.

Progression into mitosis with unresolved branched DNA structures formed from recombination intermediates can lead to the formation of ultra-fine anaphase bridges (UFBs) and ultimately DNA breaks and genome instability [19,48]. UFBs, including those formed as a result of unresolved homologous recombination joints (HR-UFBs), are coated by the PICH (PLK1-interacting checkpoint helicase) protein and can be detected in anaphase using PICH-specific antibodies [19]. It has previously been shown that depleting both the dissolvase (i.e., siBLM) and resolvase (i.e., siSLX4) branches of BIR resolution led to an increase in telomeric bridges [20]. Given that loss of RAD54 leads to an increase in the recruitment of components of the SMX resolvasome to ALT telomeres (Fig 4C and D), we hypothesized that combined depletion of RAD54 and SLX4 would lead to an increase in UFB formation. Here, we show that similar to the combined depletion of BLM and SLX4, combined depletion of RAD54 and SLX4 in ALT-positive cells leads to an increase in UFBs and an increase in the number of bridges per anaphase (Figs 4G–I, and EV4D and E). When BLM, RAD54, and SLX4 were all simultaneously depleted, there was no additional increase in bridge formation or the number of bridges per anaphase, indicating that BLM and RAD54 may be working in the same pathway to promote resolution of recombination intermediates (Fig 4G–I). Depletion of each enzyme individually did not lead to an increase in UFB formation (Fig 4G–I), likely because cells have redundant mechanisms to resolve these structures and prevent bridge formation. Perhaps more notably, we found that cells depleted for RAD54 and SLX4 had a significant increase in anaphase bridges coated by the telomere binding protein TRF2 (Fig 4J and K), demonstrating an accumulation of telomeric UFBs. Moreover, BLM binding to UFBs in the absence of RAD54 was unaffected, suggesting that the defects we observed in anaphase were not simply a product of defects in BLM-mediated resolution of UFBs (Fig EV4F and G). In addition, we did not observe significant changes in the percentage of anaphases

containing either PICH-coated bridges or BLM-coated bridges when RAD54 alone was depleted (Fig EV4H). Thus, the defects we observed in UFB resolution were not indirect effects due to loss of other known UFB resolving enzymes.

The persistence of unresolved UFBs through telophase can lead to an accumulation of fragmented DNA and the formation of micronuclei [19,49,50]. Moreover, depletion of the enzymes involved in UFB resolution, including PICH and BLM, leads to

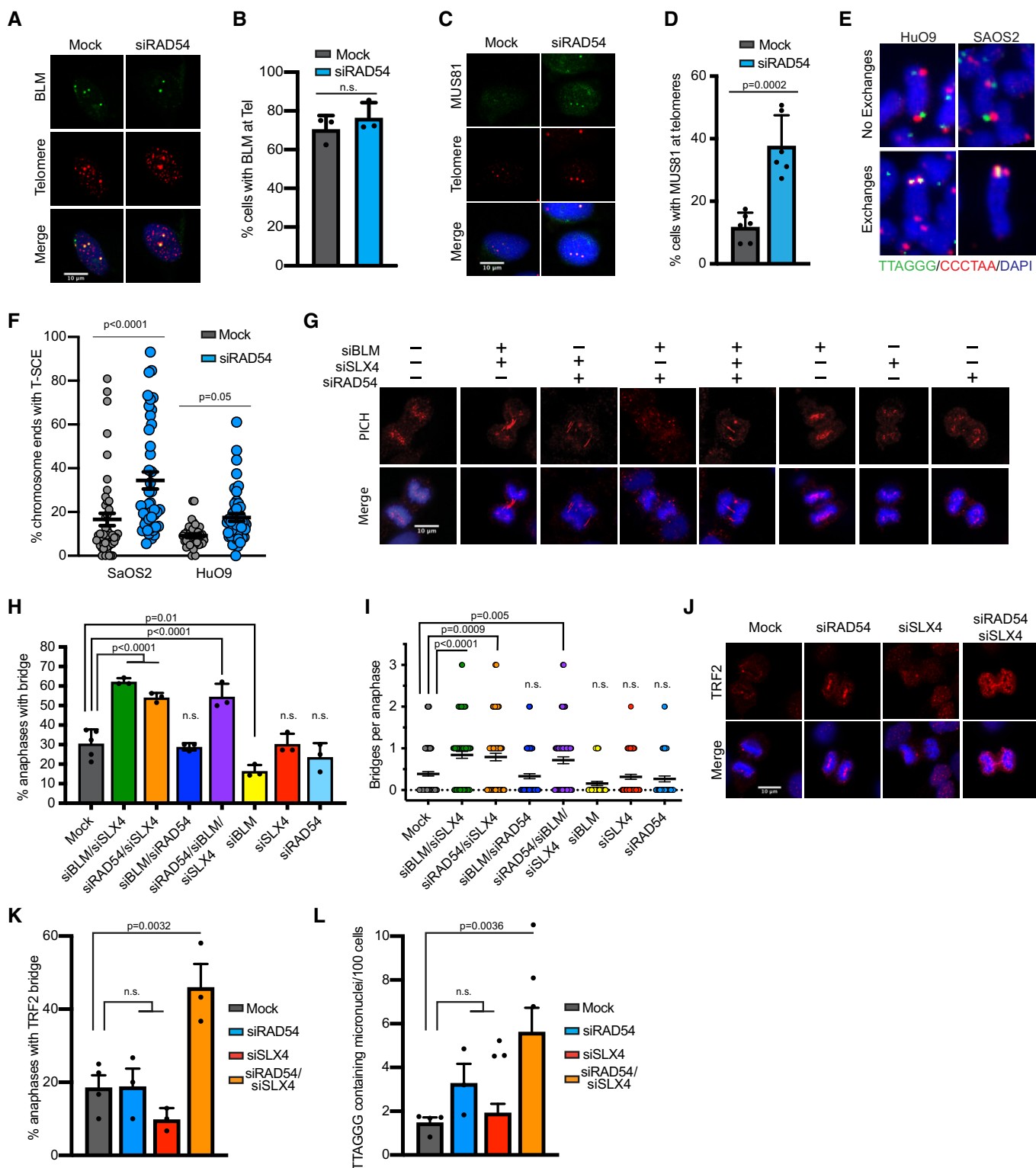

**Figure 4.**

◀

**Figure 4. RAD54 promotes post-synaptic processing of recombination intermediates at ALT telomeres.**

A   Representative images of combined IF and DNA FISH for BLM (IF) and telomeres (FISH). HuO9 cells were transfected with 20 nM siRAD54#2 for 48 h. Scale bars = 10 μm.

B   Quantification of data shown in A. Cells were counted positive if they contained at least 1 colocalization event between BLM and telomeres. Data were normalized to mock condition of each repeat. At least 100 cells per condition per repeat were counted, n = 3. Values shown are mean ± SD. Conditions were analyzed using unpaired two-tailed Student's *t*-test.

C   Representative images of IF-DNA FISH for MUS81 and telomeres. HuO9 cells were transfected with 20 nM siRAD54 #2 for 48 h prior to staining. Scale bars = 10 μm.

D   Quantification of data shown in C. Cells were counted positive if they contained at least 1 colocalization event between MUS81 and telomeres. At least 100 cells per condition were counted per repeat. n = 5, values shown are mean ± SD. Conditions were analyzed using unpaired two-tailed Student's *t*-test.

E   Representative images of HuO9 and SaOS2 chromosomes with or without an exchange event. Colocalization of green and red signal indicates an exchange event.

F   Quantification of telomere sister chromatid exchange events. Cells were transfected with 20 nM of siRNA for 48 h prior to the addition of BrdU/BrdC. Metaphases were imaged by confocal microscopy, and each data point represents a metaphase. For each condition, there were three biological repeats with a total of at least 1,300 chromosome ends quantified. Values shown are mean ± SEM. Conditions were analyzed using two-way ANOVA followed by Sidak's test.

G   Representative images of UFB detection staining for PICH in HuO9 cells. Cells were transfected with 20 nM of noted siRNAs for 72 h. Scale bars = 10 μm.

H   Quantification of data shown in G. Anaphases were scored from 4 independent experiments (mock) or 3 independent experiments (remaining conditions). Anaphases were considered positive for UFB if there was PICH staining between DAPI bodies. Single knockdown conditions had > 50 anaphases scored. Combined knockdown conditions had > 88 anaphases scored. Values shown are mean ± SD. Conditions were compared using one-way ANOVA followed by Dunnett's multiple comparison test.

I    Quantification of data shown in G. Number of PICH-stained UFBs counted per anaphase. Values shown are total counts over 3 independent experiments, shown mean ± SEM. Total number of anaphases per condition was as follows: Mock n = 134, siBLM/siSLX4 n = 90, siRAD54/siSLX4 n = 96, siRAD54/siBLM/siSLX4 n = 88. Conditions were compared using Kruskal–Wallis test followed by Dunn's multiple comparison test.

J    Representative images of UFB staining for TRF2 in HuO9 cells. Cells were transfected with 20 nM siRAD54#2 alone or cotransfected siRAD54#2 and siSLX4 for 72 h prior to staining. Scale bars = 10 μm.

K   Quantification of data shown in J. Anaphases were scored from 3 independent experiments. A total of 90 anaphases were scored per condition. Values shown are mean ± SD. Conditions were compared using one-way ANOVA followed by Dunnett's multiple comparison test.

L   Quantification of micronuclei from HuO9 cells transfected with siRAD54#2 alone or siRAD54#2 cotransfected with siSLX4 for 48 h. Micronuclei containing telomeric DNA were identified using DAPI and telomere FISH. Micronuclei were normalized to the number of nuclei counted. Values shown are mean ± SD of n = 3 biological replicates. Conditions were compared using one-way ANOVA followed by Dunnett's multiple comparison test.

persistent UFBs and increased micronuclei formation [49,50]. Likewise, when RAD54 was depleted in combination with SLX4, we observed a significant increase in the presence of micronuclei containing telomere FISH signal. These results support the hypothesis that RAD54 functions to prevent the formation of UFBs at ALT telomeres and to maintain genome stability (Figs 4L and EV4I). While we propose that the telomeric DNA found in the micronuclei is a result of fragmented telomeric UFBs, we cannot rule out the possibility that the micronuclei contain ECTRs. Given that C-circles decrease in the absence of RAD54, we do not expect that ECTRs fully explain the increase in telomeric DNA containing micronuclei observed upon RAD54 and SLX4 depletion.

Loss of RAD54 blunts elongation events at ALT telomeres in a manner dependent on the ATPase activity and oligomerization of RAD54 (Fig 3A, B, G and H). This suggests that RAD54 is promoting branch migration of recombination intermediate structures formed at ALT telomeres. Recombination intermediate structures can be resolved through dissolution or nucleolytic cleavage, or if left unresolved will lead to HR-UFBs [19]. The combined depletion of RAD54 and SLX4 led to a significant increase in UFBs at ALT telomeres (Fig 4J and K). The BLM protein has both helicase and branch migration activities and has been demonstrated to regulate ALT activity. In addition to its role in the BTR dissolvasome, BLM contributes to BIR at ALT telomeres by cooperating with DNA2 to promote long-range 5′ to 3′ resection of broken DNA ends during pre-synapsis, generating structures capable of initiating strand invasion and inducing telomere elongation events [20,44]. Several recent studies have suggested that the branch migration activity of BLM may be secondary at ALT telomeres as compared to other functions of BLM, such as its ability to recruit TOP3A-RMI1-RMI1 or promote long-range resection [20,44]. Branch migration for the dissolution of recombination intermediates at ALT telomeres may then depend on other cellular enzymes, such as RAD54 (Fig 5). This is of particular interest because RAD54 has

been shown to have higher branch migration efficiency than BLM on partial-X junctions [33,34], the DNA joints formed during BIR [36].

BIR is known to be an error-prone process and likely to introduce mismatches into telomeric DNA during POLD3 synthesis [51,52]. MUS81 has been implicated in limiting the mutagenic nature of BIR, especially at repetitive DNA elements, by cleaving D-loop structures and limiting the duration and length of extension events [51]. We observed that when RAD54 was depleted, there was an increase in MUS81 recruitment to ALT telomeres (Fig 4C and D) and a concomitant decrease in telomere synthesis events (Fig 3A and B), suggesting that RAD54 may be promoting these long-range BIR synthesis events that are otherwise terminated through MUS81 cleavage of the D-loop. Given the known interaction between RAD54 and MUS81 [39], RAD54 may function to regulate the substrate specificity of the MUS81 endonuclease at ALT telomeres [53].

In addition to favoring partial-X junctions, RAD54 promotes branch migration through sites of heterology more efficiently than BLM by an order of magnitude [36]. DNA heterology can form during strand invasion when the invading strand lacks complete homology with the template DNA. While no studies have looked specifically at the formation of DNA heterology at ALT telomeres, DNA mismatch repair proteins MSH2/6 are shown to associate with ALT telomeres [17]. The error-prone nature of BIR could, in part, account for the increase in variant telomeric repeats at ALT telomeres [54], which may lead to DNA heterology during search and capture steps of recombination. Telomeric D-loop structures containing sites of heterology may be tolerated and extended, dependent on the branch migration activity of RAD54. Alternatively, these structures may be dissolved by RAD54 prior to DNA synthesis in order to limit heterologous recombination and provide an additional opportunity for the strand invasion machinery to form a productive D-loop (Fig 5). Both possible functions of RAD54 would support *de novo* telomere synthesis and elongation events. Together, our data

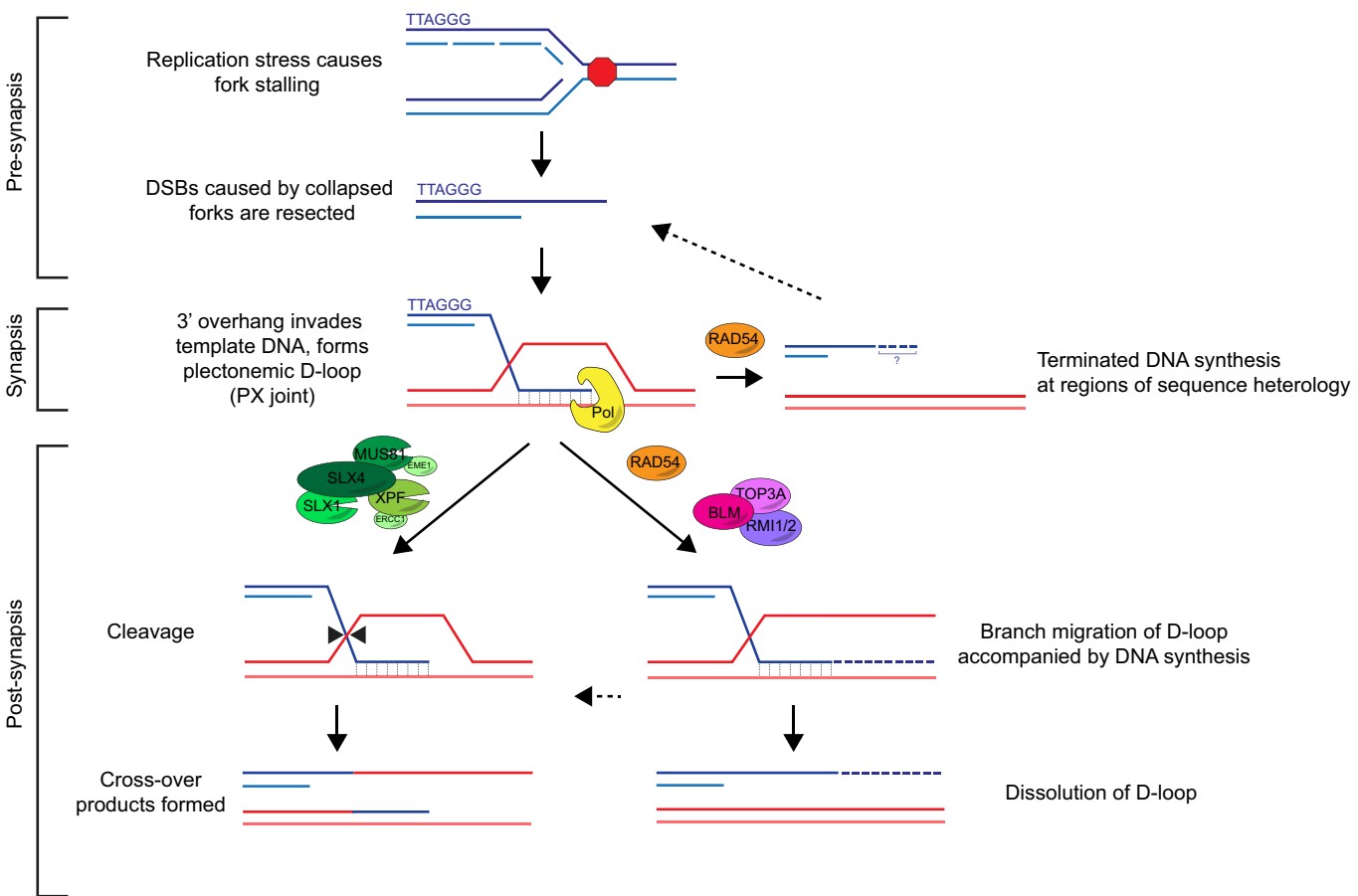

**Figure 5. Proposed model of RAD54 activity at ALT telomeres.**

RAD54 functions downstream of pre-synaptic events to regulate the processing of D-loop structures (partial-X or PX junction), promoting telomere synthesis via BIR at ALT telomeres. RAD54 branch migration activity may function coordinately with BLM to promote BTR-dependent dissolution of recombination intermediates. Specifically, RAD54 may promote branch migration and ultimately dissolution through regions of heterology either in the D-loop or that arise during error-prone DNA extension events. Alternatively, RAD54 may function to dissolve heterologous recombination intermediates to promote formation of more homologous intermediates that will lead to higher fidelity telomere extension events. In the absence of RAD54, like in the absence of BLM, ALT cells rely on nucleolytic cleavage of recombination intermediates by the SMX resolvase complex.

highlight a previously uncharacterized role for the translocase activity of RAD54 in promoting BIR-mediated telomere elongation in ALT-positive cancer cells.

## Materials and Methods

### siRNAs, cDNAs, and primers

All siRNA transfections were performed using Lipofectamine RNAiMax reagent in Opti-MEM. siRNA was mixed with RNAiMax into Opti-MEM media and incubated for 15 min at room temperature before being added to cell culture media. All plasmids were transfected using FuGENE 6 Transfection Reagent. cDNA was mixed with FuGENE 6 in Opti-MEM media and incubated for 20 min at room temperature before being added to cell culture media. Cells were plated 16–24 h before FuGENE transfection.

Polη-GFP plasmid was a generous gift from Dr. Sharon Cantor. GFP-BLM plasmid was a gift from Nathan Ellis (Addgene plasmid

#80070) N-myc-TRF2 plasmid was a gift from Titia de Lange (Addgene plasmid #16066). WT-RAD54 plasmid was a gift from Dr. Markus Lobrich and was then modified using InFusion cloning technique to introduce K189R, S49E, and silent siRNA resistance mutations as was well as to move the gene insert into an pDEST-SFB backbone.

ON-TARGETplus siRNAs were obtained from Dharmacon, siRAD54#1 (AGAAUGAUCUGCUUCACUA) and siRAD54#2 (CGAA UUACACCCAGACUUU), SLX4 (GCUACCCGGACACUUGUCAUUGU UA), and BLM (GAUCAAUGCUGCACUGCUU). siRNA for RAD51 was obtained from Ambion (UGAUUAGUGAUUACCACUG).

The following primers were used for RT-qPCR: GAPDH For (CAGAACATCATCCCTGCCTCTAC), GAPDH Rev (TTGAAGTCAG AGGAGACCACCTG), SLX4 For (TTGGTCCTACAGCGAATGCAG), and SLX4 Rev (CATGTGCCGATGCTCCTACC).

### Antibodies and probe

The following antibodies and probes were used where noted: BLM (interphase foci Abcam ab2179, UFBs Bethyl A300-110A), GAPDH

(Santa Cruz sc-47724), GFP (Abcam, ab1218), mCherry (Takara 632543), MUS81 (Santa Cruz sc53382), myc (Thermo Fisher MA1-980), PCNA (Cell Signaling Technology, 13110S), PICH (Millipore 04-1540), PML (Santa Cruz sc-5621), PML (Santa Cruz sc-966), RAD51 (Santa Cruz sc-8349, IF), RAD51 (Abcam ab176458, ChIP and Western blot), RAD54 (Santa Cruz sc-374598), TRF1 (Millipore 04-638), TRF2 (Millipore 05-521), and Tubulin (Cell Signaling Technology 2125S).

The Telomere probe (CCCTAA)4 and Alu repeat probe (GTGATCCGCCCGCCTCGGCCTCCCAAAGTG) were obtained from Invitrogen and used for dot blots where noted. The C-rich Cy3-labeled telomere probe for DNA FISH was obtained from PNA Bio (F1002).

## Cell culture

HeLa 1.2.11, U2OS, U2OS-TRF1-FOK1-WT, and U2OS-TRF1-FOK1-D450A were cultured in DMEM, 10% FBS, 1% penicillin/streptomycin. HuO9, NOS1, and SJSA1 were cultured in RPMI-1640, 5% FBS, 1% penicillin/streptomycin, 1% sodium pyruvate. Cal72 was cultured in DMEM/F12, 10% FBS, 1% penicillin/streptomycin. SaOS2 was cultured in RPMI-1640, 10% FBS, 1% penicillin/streptomycin. All cells were grown at 37°C in a humidified incubator with 5% $CO_2$. U2OS-TRF1-FOK1 cells were obtained as a gift from Dr. Roger Greenberg. These cells were treated where indicated with doxycycline (Sigma D9891), (Z)-4-hydroxytamoxifen (Sigma H7904), and Shield1 (Takara 632189). Certificate of authentication for cell lines is available upon request.

## Cell cycle analysis

Cells were harvested and washed with PBS prior to overnight fixation in 70% ethanol at −20°C. The following day, cells were pelleted and washed twice with PBS. RNA was digested with RNAse A (250 μg/ml) at 37°C for 30 min. Propidium iodide (Sigma) was diluted to 50 μg/ml in PBS and then added 1:1 to cells. Cells were incubated with propidium iodide for at least 10 min in the dark. Cells were analyzed on BD FACSCalibur. At least 10,000 events were collected. Cells were gated to remove doublets and debris. Cell cycle profile was analyzed using FlowJo v10, using the Watson pragmatic algorithm.

## Western blot

Western blots were performed using standard protocols. Briefly, samples were lysed in 2× sample buffer at 95°C for 15 min. Soluble protein lysates were run on SDS–PAGE and transferred onto PVDF membranes. Membranes were blocked in TBS-T (1× TBS, 0.1% Tween-20) containing 5% dry non-fat milk. Membranes were incubated overnight at 4°C in primary antibody diluted in 5% milk in TBS-T. Following overnight incubation, membranes were washed three times with TBS-T for 5 min each at room temperature and then incubated with horseradish peroxidase-conjugated secondary antibodies diluted in 5% milk in TBS-T. Following secondary, membranes were washed 3 × 5 min and then visualized using enhanced chemiluminescence reagents from Bio-Rad.

## Combined immunofluorescence and DNA FISH

Combined immunofluorescence and DNA FISH were performed as previously described [55]. Cells were washed twice with PBS for 5 min each. The cells were then treated with cytobuffer (100 mM NaCl, 300 mM sucrose, 3 mM $MgCl_2$, 10 mM PIPES pH 7, 0.1% Triton X-100) for 7 min at 4°C. Cells were next rinsed with PBS and fixed with 4% paraformaldehyde in PBS for 10 min at room temperature. For PCNA staining, fixation was instead performed with cold methanol at −20°C for 10 min. To permeabilize the cells, 0.5% NP40 in PBS was used for 10 min at room temperature. Following a PBS rinse, cells were blocked for 1 h at room temperature in PBG (0.5% BSA, 0.2% fish gelatin, in PBS) and then incubated with primary antibody diluted in PBG overnight at 4°C. The cells were washed three times with PBS for 5 min each at room temperature and then incubated with secondary antibody diluted in PBG for 45 min at room temperature in the dark. After this incubation, the cells were washed three times with PBS for 5 min each and then fixed with 4% paraformaldehyde in PBS for 10 min at room temperature. Cells were then digested with RNaseA 200 μg/ml in 2× SSC for 30 min at 37°C. A series of ethanol washes (70, 85, 100%) for 2 min each at room temperature was used to dehydrate the cells. The coverslips were then dried at 37°C for 20 min. Telomere probe (PNA-(CCCTAA)$_4$ Tel-Cy3) diluted 1:750 in hybridization buffer [70% formamide, 0.25% blocking reagent (Roche), 10 mM Tris pH 7.5, 4.1 mM $Na_2HPO_4$, 1.25 mM $MgCl_2$, 0.45 mM citric acid]. Slides were incubated with probe and denatured at 85°C for 3 min and then placed in a humidified chamber at 37°C overnight. Coverslips were washed with 2× SSC and formamide mixed 1:1 three times for 5 min each at 37°C, and then three times in just 2× SSC at 37°C. 1 μg/ml DAPI was added into 2× SSC for a final 10-min wash at room temperature. Coverslips were mounted on glass slides using VECTASHIELD mounting medium.

## Polη IF-FISH

Cells grown on coverglass were washed with PBS and then fixed with 4% paraformaldehyde for 30 min at room temperature. After washing with PBS, cells were permeabilized with 0.5% Triton X in PBS for 15 min. Primary antibody (GFP, abcam 1218) was diluted 1:500 in 3% BSA and 0.05% Tween-20 in PBS. Cells were stained with primary antibody overnight. Cells were washed three times for 5 min each with PBS and then incubated with secondary antibody diluted in 3% BSA and 0.05% Tween-20 in PBS for 1 h at room temperature. Following secondary antibody, the staining protocol is identical to the standard IF-DNA FISH protocol.

## Immunoprecipitation

Cells were harvested, washed, and pelleted. Pellet was lysed in NETN buffer (150 mM NaCl, 20 mM Tris pH 7.5, 0.5 mM EDTA, 0.5% NP-40, protease inhibitor) for 30 min on ice. Lysate was sonicated 30 s on/20 s off in QSonica water bath sonicator at 100% amplitude for 3 min. Lysate was centrifuged at full speed for 10 min at 4°C. Supernatant was collected. 10% of lysate was removed for input and stored at 4°C. Remaining supernatant was divided into two tubes and incubated with either 2 μg antibody or IgG control overnight at 4°C. The next morning, Protein A magnetic DynaBeads

were washed with NETN and added into antibody–lysate mixture for 1 h at 4°C. Beads were purified using a magnet and then washed three times with NETN. Inputs were treated the same for the remainder of protocol. The inputs and IP samples were boiled in 2× SDS sample buffer for 15 min at 95°C. Beads were removed with magnet, and samples were analyzed by Western blot.

### EdU incorporation and FISH

Cells were grown on coverglass and pulsed with 10 μM EdU (5-ethynyl-2′-deoxyuridine) in normal cell culture conditions for 1.5 h immediately prior to staining. Coverslips were washed with PBS and then incubated with cytobuffer (100 mM NaCl, 300 mM sucrose, 3 mM MgCl$_2$, 10 mM PIPES pH 7, 0.1% Triton X-100) for 7 min at 4°C. Following a PBS wash, cells were fixed with 4% paraformaldehyde for 10 min at room temperature. Coverslips were washed with PBS and then permeabilized with 0.5% NP40 in PBS for 10 min at room temperature. Cells were washed twice for 5 min each with PBS prior to Click-It reaction labeling. Click-it chemistry was performed according to the manufacturer's instructions. Coverslips were incubated in 10 μM Alexa Fluor 488 Azide (Thermo Fisher) diluted in 100 mM Tris pH 8.5, 1 mM CuSO$_4$, and 100 mM ascorbic acid, 30 min at room temperature in the dark. Coverslips were washed three times for 5 min each in 1× TBS + 0.2% Triton X-100 at room temperature. When combined with FISH, the FISH protocol above was followed, beginning with the RNase digestion step.

### C-circle assay

Genomic DNA was purified using QiaAMP DNA Mini Kit according to the manufacturer's instructions. DNA was digested overnight with Alu1 and Mbo1 restriction enzymes and then purified using Qiagen PCR Purification Kit. The DNA was diluted and quantified using a NanoDrop spectrophotometer. 80 ng of gDNA was diluted in 25 μl of 1× Φ29 buffer (NEB) containing BSA (NEB; 0.08 mg/ml), 0.1% Tween, 0.25 mM each dATP, dGTP, dTTP. Samples were incubated in the presence (+Φ29) or absence (−Φ29) of 7.5 U Φ29 polymerase (NEB) at 30°C for 8 h and then 65°C for 20 min.

Amplification products were boiled in 10× SSC and then run through a Bio-Rad vacuum dot blot manifold onto Hybond N+ membrane. The membrane was UV crosslinked for 35 s (125 J) and then pre-incubated in Ultra-Hyb hybridization buffer (Ambion) for 1 h at 50°C. Telomere probe (CCCTAA)$_4$ was labeled with digoxigenin using the DIG oligonucleotide 3′-end labeling kit (Roche) according to the manufacturer's instructions. Labeled probe was diluted 1:1,000 into hybridization buffer, and membrane was incubated at 50°C overnight. The membrane was washed twice, 5 min each at room temperature with 2× SSC + 0.1% SDS, and then twice 15 min each at 50°C in 0.5× SSC + 0.1% SDS. The membrane was then prepared and developed using the DIG Wash and Block Buffer Set (Roche), anti-DIG-AP (Roche), and CDP-Star (Roche) according to the manufacturer's instructions. Membrane was developed using Bio-Rad chemiluminescent imager.

### Chromatin immunoprecipitation

Cells were crosslinked in 1% formaldehyde for 8 min at room temperature. Crosslinking was stopped by adding glycine and

diluted to 0.125 M for 5 min at room temperature. Cells were washed twice with ice-cold PBS and then collected. Cells were lysed in cellular lysis buffer (5 mM PIPES, 85 mM KCl, 0.5% NP-40, protease inhibitor) for 5 min on ice and then spun down at 60 × g for 5 min. Supernatant was removed, and pellet was lysed with nuclear lysis buffer (50 mM Tris pH 8, 10 mM EDTA pH 8, 0.2% SDS, protease inhibitor) at 4°C. Chromatin samples were sonicated (30 s on/20 s off) in a QSonica water bath sonicator at 4°C for 90 min, to generate chromatin fragments between 150–500 base pairs. Sonicated chromatin samples were spun down at full speed, 4°C for 10 min, and supernatant was collected.

Chromatin samples (300 μg) were diluted ninefold in dilution IP buffer (16.7 mM Tris pH 8, 1.2 mM EDTA, 167 mM NaCl, 0.01% SDS, 1.1% Triton X-100, protease inhibitor) and incubated with antibody (4 μg) overnight at 4°C. Inputs were removed from sonicated chromatin sample and stored separately. Washed Protein A or Protein G magnetic Dynabeads were added into IP mixture for the final 2 h. Beads were washed for 3–4 min each at room temperature with the following buffers: twice with dilution IP buffer, once with TSE buffer (20 mM Tris pH 8, 2 mM EDTA pH 8, 500 mM NaCl, 1% Triton X-100, 0.1% SDS), once with LiCl buffer (100 mM Tris pH 8, 500 mM LiCl, 1% sodium deoxycholate, 1% NP-40), and twice with TE (10 mM Tris pH 8, 1 mM EDTA pH 8). Inputs were treated the same for the remainder of protocol. Samples were incubated in elution buffer (50 mM NaHCO$_3$, 140 mM NaCl 1% SDS) containing 67 μg/ml proteinase K for 1 h at 55°C. Beads were removed from eluted samples using a magnet, and the supernatants were collected, and incubated overnight at 65°C. Samples were treated with 1.2 mg/ml RNAseA for 30 min at 37°C; then, samples were cleaned up using a Qiagen PCR Purification Kit, eluting in water.

Samples were boiled for 5 min at 95°C and then diluted to 10× SSC and run through a Bio-Rad vacuum dot blot manifold onto Hybond N+ membrane to detect telomeric DNA using dot blot. The membrane was UV crosslinked for 35 s (125 J) and then pre-incubated in Ultra-Hyb hybridization buffer (Ambion) for 1 h at 50°C. Telomere probe (CCCTAA)$_4$ was labeled with digoxigenin using the DIG oligonucleotide 3′-end labeling kit (Roche) according to the manufacturer's instructions. Labeled probe was diluted 1:1,000 into hybridization buffer, and membrane was incubated at 50°C overnight. The membrane was prepared and developed using the DIG Wash and Block Buffer Set (Roche), anti-DIG-AP (Roche), and CDP-Star (Roche) according to the manufacturer's instructions. Following detection using the (CCCTAA)$_4$ telomeric probe, blot was stripped by boiling 0.1% SDS in water and incubating membrane in solution for 15 min. The membrane was then reprobed using a DIG-labeled probe specific to Alu repeats and developed using the DIG Wash and Block Buffer Set (Roche), anti-DIG-AP (Roche), and CDP-Star (Roche) according to the manufacturer's instructions.

### Detecting UFBs

Cells were grown on coverglass to 80–90% confluence; agitation was avoided as much as possible throughout the protocol. Cells were washed gently with PBS. Pre-extraction buffer A [0.2% Triton in 1× PEM buffer (20 mM PIPES pH 6.8, 1 mM MgCl$_2$, 10 mM EGTA)] was added to PBS on coverslips, diluting the buffer 1:1 in PBS. Cells were incubated for 60 s at room temperature. Pre-

extraction buffer B (0.1% Triton, 8% PFA in 1× PEM) was added directly into pre-extraction A buffer/PBS mixture, diluting pre-extraction buffer B 1:1. Cells were incubated for 15 min at room temperature. Pre-extraction buffers were removed from the cells and discarded; cells were washed three times for 5 min each with PBS. Coverslips were incubated at 4°C overnight in PBSAT (3% BSA, 0.5% Triton in 1× PBS).

Coverslips were then incubated with primary antibody diluted in PBSAT overnight at 4°C in a humidified chamber. After primary antibody, coverslips were washed with PBSAT three times for 10 min each at room temperature. Secondary antibody was diluted in PBSAT, and cells were incubated with secondary antibody for 2 h at room temperature. Coverslips were washed three times for 15 min each with PBSAT and then for 10 min with PBS. To stabilize staining, antibodies were fixed with post-staining fixation buffer (4% PFA in 1× PBS) for 5 min at room temperature. Coverslips were washed three times for 5 min each with PBS, followed by a 5-min incubation with DAPI diluted in PBS. Cells were washed a final time with PBS, followed by a rinse with $H_2O$ prior to mounting coverslips using VECTASHIELD mounting media.

**Telomere sister chromatid exchange (T-SCE)**

Cells were plated with or without siRNA for 48 h. After 48 h, 7.5 μM BrdU and 2.5 μM BrdC were added into media for 16–18 h. Cells were then treated with 20 ng/ml of colcemid for 2–4 h to enrich for mitotic cells. Cells were harvested by trypsinization and resuspended in 5× pellet volume of prewarmed 75 mM KCl and incubated at 37°C for 15 min. Ice-cold fixative (3:1 methanol/acetic acid) was gently added dropwise to cells and then gently mixed. Cells were centrifuged $146 \times g$ for 3 min at 4°C and then resuspended in fixative. This wash was repeated two more times before gently resuspending cell pellet in 500 μl of fixative. Cells were dropped onto a slide, and after approximately 20 s, the slide was held at 85°C for 3 s. Slides were allowed to air-dry for at least 48 h.

Slides were rehydrated in 2× SSC and then incubated with 0.5 mg/ml RNase A in 2× SSC for 30 min at 37°C. Slides were rinsed with PBS and then fixed with 4% paraformaldehyde for 10 min at room temperature. Slides were dehydrated through an ethanol series (75, 85, 100%) for 2 min each and then allowed to air-dry. Slides were rehydrated in 2× SSC and then incubated with 0.5 μg/ml Hoescht 33258 in 2× SSC for 15 min.

DNA was nicked by exposing the slides to 365 nm UV light. Slides were placed in the bottom of a utility box, and 2× SSC was added until just covering all the slides. The utility box containing the slides was placed on a platform in a Stratalinker, with the slides being 1–2 inches from the 365 nm bulbs. The slides were cross-linked for 45 min, making sure that the slides stayed immersed in 2× SSC for the full incubation. Following UV treatment, the slides were incubated with 10 U/μl ExoIII (Promega) for 30 min at 37°C to degrade nicked DNA. Slides were rinsed in water, denatured in 70% formamide/30% 2× SSC for 2 min at 80°C, and then immediately dehydrated through a series of ethanol washes (70, 85, 100%) for 2 min each. Slides were allowed to air-dry.

Slides were probed with TelC-Cy3 (PNA Bio) for 1.5 h at 37°C. Probe was diluted 1:750 in hybridization buffer [70% formamide, 0.25% (w/v) blocking reagent (Roche), 10 mM Tris pH 7.5, 4.24 mM $Na_2HPO_4$, 1.25 mM $MgCl_2$, 0.45 mM citric acid]. Slides

were washed three times in wash buffer A (70% formamide, 10 mM Tris pH 7.5). Slides were then probed with TelG-488 (PNA Bio) for 1.5 h at 37°C, with probe diluted 1:750 in hybridization buffer. Slides were washed three times with wash buffer A (70% formamide 10 mM Tris pH 7.5). Slides were then washed three times in wash buffer B (50 mM Tris pH 7.5, 150 mM NaCl, 0.08% Tween-20). Slides were stained with 5 μg/ml DAPI diluted in 2× SSC for 10 min at room temperature before mounting coverslips with VECTASHIELD and sealing slides with nail polish.

**RT–qPCR analysis of gene expression**

Cells were harvested, washed, and pelleted. RNA was extracted using Qiagen RNeasy Mini Kit according to the manufacturer's instructions. 500 ng of RNA was reverse-transcribed using Super-Script IV (Thermo Fisher) according to the manufacturer's instructions. cDNA was diluted 1:10 into qPCR with PowerUP SYBR Green Master Mix, and 10 μM reverse primer and 10 μM forward primer. qPCR was performed on a StepOne qPCR machine using the following program: 50°C 2 min, 95°C 2 min, and then 40 cycles (95°C 15 s, 60°C 15 s, 72°C 1 min), followed by a standard melt curve. Data were analyzed by dd_CT method and corrected to GAPDH and mock sample. Samples were run in technical triplicate.

**Statistical analysis**

Experiments were performed at least three times independently. Statistical analysis was performed using GraphPad Prism 8 software. Exact *P*-values are given for each test, and *P*-values were considered significant if they were < 0.05.

Expanded View for this article is available online.

## Acknowledgements

We would like to thank all members of the Flynn laboratory for helpful discussions and suggestions, the BUMC Flow Cytometry Core Facility, and the BUMC Cellular Imaging Core Facility, and Technical Director, Dr. Michael T. Kirber. R.L.F was supported by 1R01CA201446, an Edward Mallinckrodt Junior Foundation Award, and a Peter Paul Professorship. E.M.O was supported by NIGMS T32GM008541 and the PhRMA Foundation Pre-doctoral Fellowship in Pharmacology. N.L. was supported by the Boston University Undergraduate Research Opportunities Program. The funders had no role in study design, data collection and analysis, decision to publish, or preparation of the manuscript.

## Author contributions

EM-O designed experiments, conducted experiments, analyzed the data, and wrote the manuscript. RLF designed experiments, analyzed the data, and wrote the manuscript. NL, KT, YJL, and LMC conducted experiments and analyzed the data.

## Conflict of interest

The authors declare that they have no conflict of interest.

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
