## [Review Process File · EMBO Reports]

RAD54 promotes Alternative Lengthening of Telomeres by mediating branch migration

Emily Mason-Osann, Katherine Terranova, Nicholas Lupo, Ying Jie Lock, Lisa M. Carson, Rachel Litman Flynn

Review timeline:	Submission date:	18 October 2019
	Editorial Decision:	8 November 2019
	Revision received:	14 February 2020
	Editorial Decision:	17 March 2020
	Revision received:	19 March 2020
	Accepted:	31 March 2020

Editor: Esther Schnapp

Transaction Report:

1st Editorial Decision

8 November 2019

Thank you for the submission of your manuscript to EMBO reports. We have now received the enclosed referee reports on it.

As you will see, all referees acknowledge that the findings are interesting and should be published. However, they also point out that the data need to be strengthened and they make several suggestions for how this can be done. I think all points are reasonable and should be addressed. Please let me know if you disagree so that we can discuss this further.

I would thus like to invite you to revise your manuscript with the understanding that the referee concerns must be fully addressed and their suggestions taken on board. Please address all referee concerns in a complete point-by-point response. Acceptance of the manuscript will depend on a positive outcome of a second round of review. It is EMBO reports policy to allow a single round of major revision only and acceptance or rejection of the manuscript will therefore depend on the completeness of your responses included in the next, final version of the manuscript.

Revised manuscripts should be submitted within three months of a request for revision; they will otherwise be treated as new submissions. Please contact us if a 3-months time frame is not sufficient for the revisions so that we can discuss this further. You can either publish the study as a short report or as a full article. For short reports, the revised manuscript should not exceed 27,000 characters (including spaces but excluding materials & methods and references) and 5 main plus 5 expanded view figures. The results and discussion sections must further be combined, which will help to shorten the manuscript text by eliminating some redundancy that is inevitable when discussing the same experiments twice. For a normal article there are no length limitations, but it should have more than 5 main figures and the results and discussion sections must be separate. In both cases, the entire materials and methods must be included in the main manuscript file.

Regarding data quantification, please specify the number "n" for how many independent experiments were performed, the bars and error bars (e.g. SEM, SD) and the test used to calculate p-

values in the respective figure legends. This information must be provided in the figure legends. Please also include scale bars in all microscopy images.

1) Your manuscript contains statistics and error bars based on $n=2$ or on technical replicates. Please use scatter blots in these cases. No statistics can be calculated if $n=2$.

2) individual production quality figure files as .eps, .tif, .jpg (one file per figure).

See https://wol-prod-cdn.literatumonline.com/pb-assets/embo-site/EMBOPress_Figure_Guidelines_061115-1561436025777.pdf for more info on how to prepare your figures.

5) a complete author checklist, which you can download from our author guidelines <https://www.embopress.org/page/journal/14693178/authorguide>. Please insert information in the checklist that is also reflected in the manuscript. The completed author checklist will also be part of the RPF.

6) Please note that all corresponding authors are required to supply an ORCID ID for their name upon submission of a revised manuscript (<https://orcid.org/>). Please find instructions on how to link your ORCID ID to your account in our manuscript tracking system in our Author guidelines <https://www.embopress.org/page/journal/14693178/authorguide#authorshipguidelines>

7) We would also encourage you to include the source data for figure panels that show essential data. Numerical data should be provided as individual .xls or .csv files (including a tab describing the data). For blots or microscopy, uncropped images should be submitted (using a zip archive if multiple images need to be supplied for one panel). Additional information on source data and instruction on how to label the files are available at <https://www.embopress.org/page/journal/14693178/authorguide#sourcedata>.

8) Our journal also encourages inclusion of *data citations in the reference list* to directly cite datasets that were re-used and obtained from public databases. Data citations in the article text are distinct from normal bibliographical citations and should directly link to the database records from

which the data can be accessed. In the main text, data citations are formatted as follows: "Data ref: Smith et al, 2001" or "Data ref: NCBI Sequence Read Archive PRJNA342805, 2017". In the Reference list, data citations must be labeled with "[DATASET]". A data reference must provide the database name, accession number/identifiers and a resolvable link to the landing page from which the data can be accessed at the end of the reference. Further instructions are available at <https://www.embopress.org/page/journal/14693178/authorguide#referencesformat>

I look forward to seeing a revised version of your manuscript when it is ready. Please let me know if you have questions or comments regarding the revision.

REFeree REPORTS

Referee #1:

In this manuscript the authors investigate the role of RAD54 in the ALT mechanism. RAD54 is a member of the SWI/SNF family of DNA-dependent ATPases that has been shown to bind to D-loops and promote branch migration. ALT utilizes break-induced replication (BIR) to extend telomeric sequences, which involves telomere extension through a D-loop intermediate, thereby implicating a role for RAD54 in the ALT pathway. The authors identify that RAD54 localizes to ALT telomeres, and this localization is enhanced in response to DNA damage by camptothecin or TRF1-FokI expression. The authors present some evidence to indicate that RAD54 functions independently of RAD51 to promote telomeric DNA synthesis by BIR, rather than functioning in the formation of D-loops themselves. Analysis of RAD54 functional mutants demonstrated that RAD54's role in telomere extension is dependent on its ATPase and branch migration activities. Finally, combined depletion of RAD54 and SLX4 caused an accumulation of unresolved recombination intermediates (ultrafine anaphase bridges). Overall, this is an interesting and straightforward manuscript describing the role of RAD54 at ALT telomeres; however, I feel it is not sufficiently thorough. The mechanistic data is somewhat lacking and could be improved upon prior to publication.

1. The conclusion that RAD54 is not involved in D-loop formation is made from the data that RAD54 binds to telomeres independently of RAD51, RAD54 depletion did not prevent PCNA recruitment to telomeres and that RAD54 is not required for the recruitment of Pol η to ALT telomeres. This seems to be quite circumstantial evidence. Even taken together, I am not sure that this is sufficient to conclude that RAD54 plays no role in D-loop formation.
2. Despite RAD54 depletion causing a decrease in nascent telomeric DNA and a decrease in C-circles, the % of cells with APBs increases slightly (which the authors attribute to defects in the resolution of DNA damage). Further evidence for an "increase in stalled or irreparable recombination intermediates" in APBs is required. APBs are typically considered to be sites of telomere extension, but the data are not consistent with this. This needs further clarification regarding what is happening in the APBs.
3. Quantitating the % of cells with APBs is unusual. I would prefer to see the number of APBs/cell. Also, does the intensity of telomeric DNA in APBs change and is there any evidence of increased DNA damage (in APBs) following RAD54 depletion.
4. Does RAD54 depletion cause a change in telomere length? Presumably you should see telomere shortening. There is currently no telomere length measurement. This should be included. Similarly, the frequency of telomere signal free ends could be quantitated.

5. The authors observe that RAD54 depletion results in decreased BLM recruitment to ALT telomeres. This requires further investigation. Is RAD54 involved in recruitment of BLM and is this what is integral to RAD54's activity at ALT telomeres. Is the increase in SMX recruitment a direct result of the decrease in BLM recruitment? Why was MUS81 investigated rather than SLX4, when SLX4 is the major player in ALT telomere resolution?
6. Does the increase in SMX (MUS81) recruitment result in an increase in T-SCEs?
7. Please check the reference formatting. Some references are duplicated.
8. Fig 4K - the authors quantitate TTAGGG-containing micronuclei. What about the actual numbers of micronuclei?
9. The siBLM/siRAD54 control should be included in Fig 4G.
10. Fig 4H, J and K should include additional controls (at least siRAD54, siRAD54/siSLX4, siSLX4).

Referee #2:

This manuscript proposes that RAD54 promotes alternative lengthening of telomeres (ALT) by mediating branch migration.

RAD54 is a DNA-dependent ATPase that translocates along dsDNA and possesses branch-migrating activity. Through a series of well-performed experiments, the authors show that RAD54 plays a role in branch migration in ALT.

Overall, I find the manuscript suitable for publication in EMBO Reports. I tend to agree with the conclusions of the authors, although I would also mention some other less likely possibilities (for example, RAD54 acting upstream of D-loop formation; see point 4, below).

The only new experiment the authors could do is some controls (point 7), although one can also live without these controls, since the controls were done in the previous experiments (point 6).

Specific Comments:

1. Fig. 1. The authors demonstrate that RAD54 colocalizes with telomeric DNA in ALT cells, mostly at ALT-associated PML bodies (APBs). The authors then induced DSBs at telomeric DNA with a chimeric TRF1-FOK1 protein. This resulted in a significant increase in the percentage of cells containing 8 or more RAD54 foci that colocalized with telomeric DNA, arguing that RAD54 localizes to telomeres that have DNA damage. All these results are well documented.
2. Fig. 2A-E. The authors show that depletion of RAD54 by siRNA did not lead to a significant change in localization of RAD51 to ALT telomeres. Depletion of RAD51 also did not affect localization of RAD54 at ALT telomeres. The results are clear, but RAD52 may be more important than RAD51 in the ALT pathway. Can the authors examine whether RAD54 affects RAD52 recruitment and vice versa?
3. Fig. 2F-G. The authors use Pol-eta-GFP as a surrogate for the formation of D-loops. They then show that loss of RAD54 did not lead to a significant change of GFP-positive cells that contained colocalization of Pol-eta-GFP with ALT telomeres. I am not sure how to interpret these data. It is a small part of the manuscript and probably could be moved to the supplementary section without compromising the manuscript. It also somewhat contradicts the results shown in Fig. 3B.
4. Fig. 3A-F. The authors show that EdU incorporation at ALT telomeres is reduced after RAD54 depletion. C-circles were also reduced, while APBs were somewhat increased. These results support a role of RAD54 in ALT, but the experiments shown in Fig. 2, do not exclude the possibility that RAD54 acts upstream of the DNA synthesis phase, since RAD52 may be more important than RAD51 for pre-synapsis and Pol-eta may not be a good surrogate for the formation of D-loops. Perhaps, the authors can mention this in the Discussion.
5. Fig. 3G-H. The authors show that wild-type RAD54 can rescue DNA synthesis, whereas ATPase-defective and oligomerization-defective mutants do not. These results are clear.

6. Fig. 4A-E. The authors show that RAD54 depletion leads to significant increase in MUS81 foci at telomeres, suggesting that in the absence of RAD54, MUS81 resolves recombination intermediates. Indeed, if RAD54 mediates branch migration in BIR/ALT, as the authors propose, then this would be the expected result.

7. Fig. 4F-K. The authors show that depletion of RAD54 and SLX4 (which functions with MUS81) leads to anaphase bridges that can contain TRF2 and telomeric repeats. These results again support the model that RAD54 promotes branch migration. Panels J and K lack an siSLX4 only control.

8. Fig. 5 shows the model of the authors. However, the proposed function of RAD54 is not clearly depicted here. Where does RAD53 function in branch migration? Which is the partial X structure that the authors refer to?

9. The characterization of ALT as a BIR process was also reported by Roumelioti et al, 2016.

Referee #3:

Alternative Lengthening of Telomeres is a recombination-based mechanism of telomere elongation used in ~10% of cancer cells. Characterization of the molecular mechanisms involved in ALT is therefore critical to develop therapies that target ALT cancers. Although multiple functions of RAD54 have been described in homologous recombination and BIR, its potential functions in ALT remained so far uncharacterized. The present manuscript from E Mason-Osann and R Flynn describe the role of RAD54 in ALT cells. First, the authors found that RAD54 is recruited to telomeres in ALT cells but not in telomerase positive cancer cells (Tel+); more specifically, Rad54 colocalizes with ALT-associated PML bodies, which are thought to serve as recombination platforms for telomeres. Because Rad54 can regulate Rad51 nucleofilaments formation during HR, the authors then tested the role of Rad54 in Rad51 foci formation in ALT, and found that Rad54 is dispensable for this step of ALT recombination. Similarly, they found that recruitment of Pol eta to APBs was not compromised after Rad54 knockdown. While not required for synapsis, Rad54 is critical for ALT mechanism: the authors found that Rad54 knockdown leads to a reduction of long-range synthesis and a decrease of C-circles. Conversely, siRad54 led to an increase in APBs, suggesting a defect in resolution of the recombination intermediates at telomeres. Recombination intermediates between ALT telomeres can be resolved by either BLM-TOP3-RMI1/2 or by SLX1/4-MUS81-XPF/ERCC1. The authors finally investigated the role of Rad54 in branch migration and recombination resolution within these two pathways, and found that 1) a rad54 mutant for branch migration cannot rescue BIR at telomeres and 2) siRAD54/siSLX4 double knockdown leads to increased telomeric bridges, suggesting that RAD54 promotes recombination intermediates resolution through the BLM pathway.

The findings are novel and of broad interest to Embo Reports' audience.

A few experiments are however required to strengthen the findings:

Major points:

1- Most of this study is based on immunofluorescence experiments and colocalization of different proteins with APBs. However, due to the low number of APBs per cell, the authors base all their findings on a very low number of colocalizations. To strengthen their claims, they need to either use an alternative approach, or induce more APB/telomere-HR protein colocalization using the TRF1-Fok1 system. In summary, the authors need to count the colocalizations shown in Figures 2a, 2b, 4c and 4d after TRF1-Fok1 induction.

2- Figure 3G-H: the authors show that there is less long-range DNA synthesis at telomeres upon loss of Rad54 branch migration activity. This is a major finding of this study. However, these conclusions again rely solely on the count for EdU colocalizations with telomeres, which are rather rare. The conclusions need to be strengthened by a different approach, such as measurement of BrdU incorporation at telomeres upon TRF1-Fok1 induction.

3- Figure 4: The authors show that Rad54 also participates in BLM-dependent resolution of recombination intermediates. They show that siRAD54/siSLX4 phenocopies the telomeric anaphase bridges observed in siBLM/siSLX4, and that triple siRAD54/siBLM/siSLX4 does not increase the phenotype. They need, however, to show that there is no increased anaphase bridges with

siBLM/siRAD54.

Minor points:

- Introduction, first sentence: "cancer cells must maintain telomere length in order to escape cellular senescence". It is rather cellular crisis.

- Introduction, paragraph 6: "Unlike other branch migration enzymes, such as BLM and REQ1, RAD54...". Do the authors mean RECQ1?

- Figure 3C: C-circle analysis. Is there a mistake in the probe used? Shouldn't it be (TTAGGG)₄?

1st Revision - authors' response

14 February 2020

Referee #1:

In this manuscript the authors investigate the role of RAD54 in the ALT mechanism. RAD54 is a member of the SWI/SNF family of DNA-dependent ATPases that has been shown to bind to D-loops and promote branch migration. ALT utilizes break-induced replication (BIR) to extend telomeric sequences, which involves telomere extension through a D-loop intermediate, thereby implicating a role for RAD54 in the ALT pathway. The authors identify that RAD54 localizes to ALT telomeres, and this localization is enhanced in response to DNA damage by camptothecin or TRF1-FokI expression. The authors present some evidence to indicate that RAD54 functions independently of RAD51 to promote telomeric DNA synthesis by BIR, rather than functioning in the formation of D-loops themselves. Analysis of RAD54 functional mutants demonstrated that RAD54's role in telomere extension is dependent on its ATPase and branch migration activities. Finally, combined depletion of RAD54 and SLX4 caused an accumulation of unresolved recombination intermediates (ultrafine anaphase bridges). Overall, this is an interesting and straightforward manuscript describing the role of RAD54 at ALT telomeres; however, I feel it is not sufficiently thorough. The mechanistic data is somewhat lacking and could be improved upon prior to publication.

1. The conclusion that RAD54 is not involved in D-loop formation is made from the data that RAD54 binds to telomeres independently of RAD51, RAD54 depletion did not prevent PCNA recruitment to telomeres and that RAD54 is not required for the recruitment of Pol η to ALT telomeres. This seems to be quite circumstantial evidence. Even taken together, I am not sure that this is sufficient to conclude that RAD54 plays no role in D-loop formation.

This is a fair point and we agree that our conclusions may have inadvertently been overstated. It's clear that *in vitro* RAD54 promotes synapsis and facilitates D-loop formation. However, several other proteins have also been reported to not only regulate synapsis and/or D-loop formation, but also regulate ALT activity including, RAD51AP1, PALB2, and HOP2-MND1 suggesting that additional recombination factors may compensate for RAD54 loss at ALT telomeres (Verma et al., 2019, Genes Dev; Garcia-Exposito et al 2016 Cell Rep). Therefore, while RAD54 may in fact mediate synapsis at ALT telomeres, it is not essential for maturation of the D-loop. We have adjusted the text to include this possibility, thank you.

2. Despite RAD54 depletion causing a decrease in nascent telomeric DNA and a decrease in C-circles, the % of cells with APBs increases slightly (which the authors attribute to defects in the resolution of DNA damage). Further evidence for an "increase in stalled or irreparable recombination intermediates" in APBs is required. APBs are typically considered to be sites of telomere extension, but the data are not consistent with this. This needs further clarification regarding what is happening in the APBs.

We agree with the reviewer that APBs are believed to be platforms for telomere recombination and/or extension events. However, this is difficult to demonstrate directly and to our knowledge there are no assays to directly demonstrate the kinetics of recombination intermediates within APB. Our data demonstrate that following RAD54 depletion APB are modestly, yet significantly, increased while EdU incorporation is significantly decreased. We have interpreted this data to mean that while the early steps in recombination (APB) are unaffected, the later steps (productive elongation) are impaired. This is supported by Response to Review Figure 5 and Figure 2 demonstrating that depletion of RAD54 did not affect the localization of RAD51 nor RAD52 to ALT telomeres suggesting that early steps in recombination are unperturbed. Moreover, previous literature demonstrated that RAD54 is required to promote DNA synthesis at a fully formed D-loop highlighting a role for RAD54 in later stages of recombination.

3. Quantitating the % of cells with APBs is unusual. I would prefer to see the number of APBs/cell. Also, does the intensity of telomeric DNA in APBs change and is there any evidence of increased DNA damage (in APBs) following RAD54 depletion.

We quantify percentage of cells positive for APB simply because we are quantifying a single plane, not Z-stacks. In this way, the number of foci per cell is limited and somewhat misleading to readers. We also find that the percentage of foci per cell will not allow the reader to appreciate the magnitude of the effect in a population of cells. For example, in a field of 10 cells if 1 cell had 10 foci we would quantify that as 10% of cells had 10 foci or greater. However, if we were to quantify by APBs/cell we would represent that as 1 APB/cell and this isn't entirely accurate given the above example. That being said, we have quantified the data

as APBs per cell and included this in Figure EV3. We do not see an increase in markers of DNA damage including, γ H2AX and 53BP1 at ALT telomeres following depletion of RAD54 (Response to Review Figure 1).

Response to Review Figure 1: HuO9 cells either mock transfected or transfected with siRAD54 and analyzed by IF-FISH using 53BP1 (A-B) or γ H2AX (C-D) antibodies and a telomere specific probe. Values shown are mean \pm sem, n=3 for γ H2AX n=2 for 53BP1 with at least 100 cells quantified per repeat. Values in A) and C) were compared using a Mann-Whitney U test

4. Does RAD54 depletion cause a change in telomere length? Presumably you should see telomere shortening. There is currently no telomere length measurement. This should be included. Similarly, the frequency of telomere signal free ends could be quantitated.

All of our experiments were performed using two different siRNA for analysis. In order to analyze telomere length over time we would need to do population double assays while also continuously transfecting cells with siRNA to ensure knockdown. We did try this with the siRNA, however, the cells were incredibly unhappy and ultimately stopped growing. We don't believe this has as much to do with RAD54 knockdown as it does repeated transfections. To overcome this technically, we generated CRISPR KO clones and analyzed telomere length by TRF (Response to Review Figure 2A-B). Again, there were challenges in generating a RAD54 KO line and while we were able to grow out single clones, it took 6 months likely allowing for adaptation to RAD54 loss. Consequently, we did not detect changes in telomere length by TRF assay using four distinct RAD54 KO clones. However, this is not unlike other genes believed to regulate ALT activity. For example, knockdown of the BTR complex has not been demonstrated to lead to telomere attrition in ALT cells (Sobinoff et al., 2017, EMBO). Likewise, RAD52 KO demonstrates very modest changes in ALT telomere length (Verma et al., 2019, Genes Dev). Given that many recombination factors have overlapping functions it is not surprising that alternative telomere recombination mechanisms can compensate to maintain ALT telomere length (Verma et al., 2019, Genes Dev).

Response to Review Figure 2: A) Western blot confirming CRISPR-mediated RAD54 knockout in HuO9 clones. B) Terminal Restriction Fragment (TRF) analysis of RAD54 KO clones.

5. The authors observe that RAD54 depletion results in decreased BLM recruitment to ALT telomeres. This requires further investigation. Is RAD54 involved in recruitment of BLM and is this what is integral to RAD54's activity at ALT telomeres. Is the increase in SMX recruitment a direct result of the decrease in BLM recruitment? Why was MUS81 investigated rather than SLX4, when SLX4 is the major player in ALT telomere resolution?

We are grateful to the reviewer for asking us to follow up on BLM recruitment. The literature does not support detection of BLM by standard IF and as a result we relied on the expression of a GFP-tagged BLM protein for analysis. While in review, we followed up on these initial experiments analyzing endogenous BLM protein using newly purchased antibodies. The new BLM antibody is able to cleanly detect endogenous BLM (Response to Review Figure 3) and after repeating the above experiments we saw no changes in BLM localization following RAD54 knockdown. This data has been revised in the main text (Figure 4).

In the absence of RAD54-mediated branch migration the SMX dissolvosome is recruited to telomeres to resolve unproductive intermediates. MUS81 is a central component of the SMX resolvosome and believed to provide specificity during nucleolytic cleavage, we analyzed MUS81 here because the SLX4 antibodies are not reliable for IF. However, we acknowledge the importance of SLX4 in the remainder of the manuscript and return our focus to SLX4 in Figure 4.

Response to Review Figure 3: Validation of BLM antibody. U2OS cells were treated with either a control or BLM siRNA and analyzed by IF-FISH using a BLM antibody and a telomere specific probe

6. Does the increase in SMX (MUS81) recruitment result in an increase in T-SCEs?

Yes. In the revised manuscript we now provide data in Figure 4 demonstrating that RAD54 depletion leads to a significant increase in T-SCE in two separate ALT cell lines supporting our model that RAD54 functions specifically to promote ALT telomere extension events.

7. Please check the reference formatting. Some references are duplicated.

Thank you for catching this, we have deleted the duplicates.

8. Fig 4K - the authors quantitate TTAGGG-containing micronuclei. What about the actual numbers of micronuclei?

The actual number of micronuclei are not statistically different amongst conditions (Response to Review Figure 4).

9. The siBLM/siRAD54 control should be included in Fig 4G.

We have now included this condition in Figure 4.

10. Fig 4H, J and K should include additional controls (at least siRAD54, siRAD54/siSLX4, siSLX4).

We have added the requested controls to Figures 4.

Referee #2:

This manuscript proposes that RAD54 promotes alternative lengthening of telomeres (ALT) by mediating branch migration.

RAD54 is a DNA-dependent ATPase that translocates along dsDNA and possesses branch-migrating activity.

Through a series of well-performed experiments, the authors show that RAD54 plays a role in branch migration in ALT.

Overall, I find the manuscript suitable for publication in EMBO Reports. I tend to agree with the conclusions of the authors, although I would also mention some other less likely possibilities (for example, RAD54 acting upstream of D-loop formation; see point 4, below).

The only new experiment the authors could do is some controls (point 7), although one can also live without these controls, since the controls were done in the previous experiments (point 6).

Specific Comments:

1. Fig. 1. The authors demonstrate that RAD54 colocalizes with telomeric DNA in ALT cells, mostly at ALT-associated PML bodies (APBs). The authors then induced DSBs at telomeric DNA with a chimeric TRF1-FOK1 protein. This resulted in a significant increase in the percentage of cells containing 8 or more RAD54 foci that colocalized with telomeric DNA, arguing that RAD54 localizes to telomeres that have DNA damage. All these results are well documented.

Thank you

2. Fig. 2A-E. The authors show that depletion of RAD54 by siRNA did not lead to a significant change in localization of RAD51 to ALT telomeres. Depletion of RAD51 also did not affect localization of RAD54 at ALT telomeres. The results are clear, but RAD52 may be more important than RAD51 in the ALT pathway. Can the authors examine whether RAD54 affects RAD52 recruitment and vice versa?

We agree with the reviewer that RAD52 may be more critical in the regulation of ALT activity, however, RAD54 has not been previously shown to regulate RAD52, whereas there have been numerous previous connections to RAD51. This was the primary reason for our investigation of RAD51 and not RAD52. That being said, we have analyzed the dependency of RAD52 on RAD54, and vice versa (Response to Review Figure 5). We saw no change in RAD52 localization following RAD54 knockdown. However, we did detect a modest increase in RAD54 localization following RAD52 knockdown. This is in agreement with the literature demonstrating that while RAD52 promotes ALT activity it is not essential and that other recombination factors compensate for RAD52 loss.

3. Fig. 2F-G. The authors use Pol-eta-GFP as a surrogate for the formation of D-loops. They then show that loss of RAD54 did not lead to a significant change of GFP-positive cells that contained colocalization of Pol-eta-GFP with ALT telomeres. I am not sure how to interpret these data. It is a small part of the manuscript and probably could be moved to the supplementary section without compromising the manuscript. It also somewhat contradicts the results shown in Fig. 3B.

We appreciate that this negative data may not seem overly important, but we feel that it helps delineate a significant role for RAD54 post synapsis. The current literature suggests that the DNA polymerases can only bind to mature plectonemic structures. The fact that we don't detect any changes in Pol η following RAD54 knockdown suggest that there is no problem in synapsis at telomeric DNA. Perhaps the more interesting piece of the data is that while Pol η is physically present at the telomeres it is unable to promote telomeric DNA synthesis which supports our hypothesis that RAD54 branch migration activity is crucial to productive elongation events.

4. Fig. 3A-F. The authors show that EdU incorporation at ALT telomeres is reduced after RAD54 depletion. C-circles were also reduced, while APBs were somewhat increased. These results support a role of RAD54 in ALT, but the experiments shown in Fig. 2, do not exclude the possibility that RAD54 acts upstream of the DNA synthesis phase, since RAD52 may be more important than RAD51 for pre-synapsis and Pol-eta may not be a good surrogate for the formation of D-loops. Perhaps, the authors can mention this in the Discussion.

Reviewer 1 also mentioned the importance of RAD52 in the ALT mechanism and we certainly agree with this point. We had focused our attention on RAD51 simply because historically speaking RAD51 and RAD54 have

consistently been hypothesized to function coordinately in pre-synapsis and this role for RAD52 has been less clear. However, given the importance of RAD52 in ALT we have analyzed the dependence of RAD54 and RAD52 on one another (See Response to Review Figure 5 above).

Response to Review Figure 5: RAD54 and RAD52 recruitment are independent of one another. A-C) HuO9 cells were either mock treated or treated with siRAD52 and analyzed by IF-FISH using a RAD54 antibody and a telomere specific probe. Values shown are mean \pm sem, n=3 with at least 100 cells quantified per repeat. Values were compared using unpaired two-tailed student's t-test. D-F) SaOS2 cells were transfected with GFP-RAD52 and then either mock treated or treated with siRAD54. Cells were analyzed by IF-FISH using a GFP antibody and a telomere specific probe. Values shown are mean \pm sd, n=3 with at least 100 cells quantified per repeat. Values were compared using unpaired two-tailed student's t-test.

5. Fig. 3G-H. The authors show that wild-type RAD54 can rescue DNA synthesis, whereas ATPase-defective and oligomerization-defective mutants do not. These results are clear.

Thank you

6. Fig. 4A-E. The authors show that RAD54 depletion leads to significant increase in MUS81 foci at telomeres, suggesting that in the absence of RAD54, MUS81 resolves recombination intermediates. Indeed, if RAD54 mediates branch migration in BIR/ALT, as the authors propose, then this would be the expected result.

Thank you

7. Fig. 4F-K. The authors show that depletion of RAD54 and SLX4 (which functions with MUS81) leads to anaphase bridges that can contain TRF2 and telomeric repeats. These results again support the model that RAD54 promotes branch migration. Panels J and K lack an siSLX4 only control.

We have added in siSLX4 as a control as requested

8. Fig. 5 shows the model of the authors. However, the proposed function of RAD54 is not clearly depicted here. Where does RAD53 function in branch migration? Which is the partial X structure that the authors refer to?

We propose the RAD54 could function in several places within the ALT mechanism, much like what has been predicted for RAD54 during homologous recombination of DNA double-strand breaks. We propose that

RAD54 branch migration activity may function coordinately with BLM to promote BTR-dependent dissolution of recombination intermediates. The partial X structure which is formed when the 3' ssDNA invades the duplex DNA to form the D-loop. Given that RAD54 has been demonstrated to have high affinity for this partial X structure we hypothesize that RAD54 facilitates the branch migration at this structure to promote telomeric DNA synthesis.

9. The characterization of ALT as a BIR process was also reported by Roumelioti et al, 2016.

We have made sure to include this citation, thank you

Referee #3:

Alternative Lengthening of Telomeres is a recombination-based mechanism of telomere elongation used in ~10% of cancer cells. Characterization of the molecular mechanisms involved in ALT is therefore critical to develop therapies that target ALT cancers. Although multiple functions of RAD54 have been described in homologous recombination and BIR, its potential functions in ALT remained so far uncharacterized. The present manuscript from E Mason-Osann and R Flynn describe the role of RAD54 in ALT cells. First, the authors found that RAD54 is recruited to telomeres in ALT cells but not in telomerase positive cancer cells (Tel+); more specifically, Rad54 colocalizes with ALT-associated PML bodies, which are thought to serve as recombination platforms for telomeres. Because Rad54 can regulate Rad51 nucleofilaments formation during HR, the authors then tested the role of Rad54 in Rad51 foci formation in ALT, and found that Rad54 is dispensable for this step of ALT recombination. Similarly, they found that recruitment of Pol eta to APBs was not compromised after Rad54 knockdown. While not required for synapsis, Rad54 is critical for ALT mechanism: the authors found that Rad54 knockdown leads to a reduction of long-range synthesis and a decrease of C-circles. Conversely, siRad54 led to an increase in APBs, suggesting a defect in resolution of the recombination intermediates at telomeres. Recombination intermediates between ALT telomeres can be resolved by either BLM-TOP3-RMI1/2 or by SLX1/4-MUS81-XPF/ERCC1. The authors finally investigated the role of Rad54 in branch migration and recombination resolution within these two pathways, and found that 1) a rad54 mutant for branch migration cannot rescue BIR at telomeres and 2) siRAD54/siSLX4 double knockdown leads to increased telomeric bridges, suggesting that RAD54 promotes recombination intermediates resolution through the BLM pathway.

The findings are novel and of broad interest to Embo Reports' audience.

A few experiments are however required to strengthen the findings:

Major points:

1- Most of this study is based on immunofluorescence experiments and colocalization of different proteins with APBs. However, due to the low number of APBs per cell, the authors base all their findings on a very low number of colocalizations. To strengthen their claims, they need to either use an alternative approach, or induce more APB/telomere-HR protein colocalization using the TRF1-Fok1 system. In summary, the authors need to count the colocalizations shown in Figures 2a, 2b, 4c and 4d after TRF1-Fok1 induction.

We appreciate this sentiment and analyzed RAD51 and MUS81 foci formation following RAD54 knockdown and TRF1-FOK1 induction. While TRF1-FOK1 induction led to a significant increase in RAD51 and MUS81 colocalization with telomeric DNA, there was no additional increase in colocalization events upon loss of RAD54. This has been added to Figure EV2 and EV4.

2- Figure 3G-H: the authors show that there is less long-range DNA synthesis at telomeres upon loss of Rad54 branch migration activity. This is a major finding of this study. However, these conclusions again rely solely on the count for EdU colocalizations with telomeres, which are rather rare. The conclusions need to be strengthened by a different approach, such as measurement of BrdU incorporation at telomeres upon TRF1-Fok1 induction.

We appreciate the reviewers concern over EdU incorporation being the only assay used to demonstrate defects in long-range DNA synthesis following RAD54 knockdown. As suggested, we attempted to validate this finding using additional assays. First, we attempted to analyze BrdU incorporation by ChIP following TRF1-FOK1 induction in RAD54 depleted cells. Although we preformed this assay multiple times using both

POLD3 and BLM siRNA as positive controls we were unable to see any decrease in BrdU incorporation following TRF1-FOK1 induction (Response to Review Figure 6). Therefore, we attempted to bolster our conclusion from a different angle. If defects in RAD54 led to a decrease in long-range synthesis at ALT telomeres than similar to loss of BLM, loss of RAD54 would lead to an increase in telomere sister-chromatid exchange, T-SCE. In the revised manuscript we now demonstrate that loss of RAD54 leads to a significant increase in T-SCE in two ALT positive cell lines (Figure 4) further supporting our initial conclusion.

Response to Review Figure 6: BrdU ChIP in TRF1-FOK1 U2OS cells. A-B) Cells were either mock transfected or transfected with siRNA for RAD54 or BLM and TRF1-FOK1 cleavage was induced using 4OHT and Shield1. Cells without 4OHT and Shield1 or without BrdU were used as negative controls. A-B) Cells were either mock transfected or transfected with siRNA for RAD54 or BLM and TRF1-FOK1 cleavage was induced using 4OHT and Shield1. Cells without 4OHT and Shield1 or without BrdU were used as negative controls. BLM and POLD3 experiments were performed in duplicate and representative images are shown

3- Figure 4: The authors show that Rad54 also participates in BLM-dependent resolution of recombination intermediates. They show that siRAD54/siSLX4 phenocopies the telomeric anaphase bridges observed in siBLM/siSLX4, and that triple siRAD54/siBLM/siSLX4 does not increase the phenotype. They need, however, to show that there is no increased anaphase bridges with siBLM/siRAD54.

We have now included the siBLM/siRAD54 bridges in our analysis (Figure 4)

Minor points:

- Introduction, first sentence: "cancer cells must maintain telomere length in order to escape cellular senescence". It is rather cellular crisis.

Yes, thank you, we have adjusted the text accordingly.

- Introduction, paragraph 6: "Unlike other branch migration enzymes, such as BLM and REQ1, RAD54...". Do the authors mean RECQ1?

Yes, we did, we have fixed the typo, thank you.

- Figure 3C: C-circle analysis. Is there a mistake in the probe used? Shouldn't it be (TTAGGG)₄?

C-circles are C-rich, following the RCA reaction with Phi29 polymerase, the amplification product generated is the TTAGGG sequence. To detect this amplification product we use a CCCTAA probe. The probe listed in the figure is correct.

2nd Editorial Decision

17 March 2020

Thank you for the submission of your revised manuscript. We have now received the enclosed comments from all referees. As you will see, referee 2 still has a few concerns that I would like you to address before we can proceed with the official acceptance of your study.

Apart from that, only a few minor changes are also required:

Please send us a full conflict of interest statement.

I attach to this email a related manuscript file with comments by our data editors. Please address all comments in the final manuscript file.

The text in the synopsis image is rather small at the final image size (550 pixels x 200 pixels). Can you may be send us a new image with bigger text? Preferably at the final image size. Thank you.

I look forward to seeing a final version of your manuscript as soon as possible. Please let me know if you have any questions or comments.

REFeree REPORTS

Referee #1:

The authors have done an excellent job responding to the reviewers' comments. I am satisfied with the responses to my concerns and the amendments/inclusions that have been made. I support publication in EMBO reports.

Referee #2:

The authors have provided a detailed response to the comments of the reviewers. However, certain elements have raised concerns and I think that the manuscript needs another round of revision. Overall, this is an interesting study and the manuscript, once revised, should merit publication in EMBO Reports.

Specific points:

1. Fig. 1A shows that a small number of telomeres (about 2 per cell) stain positive for Rad54. Similarly, Fig. 2A shows that only a small number of telomeres (about 2 per cell) stain positive for Rad51. Finally, Fig. 3A shows that only a small number of telomeres (about 2 per cell) stain positive for EdU incorporation. Presumably, it is the same telomeres that stain positive for Rad54, Rad51 and EdU incorporation, but the authors have not demonstrated this.

One of the other reviewers raised the legitimate concern that the small number of Edu-positive foci per cell makes it difficult to establish that the number of these foci decreases after Rad54 depletion, since small numbers are associated statistically with higher errors of measurement. This concern is valid and, in fact, the authors reported in the original manuscript that the number of BLM foci decreased after Rad54 depletion, but now they repeated the experiment and report that the number of BLM foci does not decrease. The authors attribute the different result to the method used (antibodies to the endogenous protein versus foci of GFP-BLM), but it is also likely that the different result is due to statistics (large variances due to small number of foci measured).

The key result of the manuscript is shown at Fig. 3b, which shows decreased EdU incorporation at telomeres after RAD54 siRNA. However, the decrease is in the range of 30-40% in 3 of the 4 experiments shown. Given the small number of EdU foci, one begins to wonder if more robust data should be collected. The authors demonstrate in Fig. 1F that expression of TRF1-FOK1 makes all

telomeres positive for RAD54 foci. Therefore, expression of TRF1-FOK1 increases dramatically the number of RAD54 foci per cell. I suppose that all the telomeres also become EdU positive, BLM positive and RAD51 positive. Therefore, this is a great system to determine whether RAD54 depletion suppresses EdU incorporation. But the authors did not use this system. Why not?

Therefore, I think that the authors should perform this one experiment, which would allow them to determine with much greater confidence if RAD54 is needed for EdU incorporation at telomeres.

2. In Fig. 2F the authors show the presence of Poleta-GFP foci at telomeres. Presumably, these foci indicate synaptic events at telomeres and one would expect about two foci per cell (same number as the RAD54 foci). This is important, because the authors use Poleta-GFP foci as a surrogate marker for synapsis and accordingly place the function of RAD54 downstream of synapsis. However, Fig. 2F shows that the majority of telomeres (or even all telomeres) stain positive for Poleta-GFP. Therefore, Poleta-GFP may not be a marker of synapsis. Can the authors please comment about this?

Referee #3:

Dr. Flynn and colleagues have performed the experiments I asked for and, together with the response to the other reviewers, have submitted a significantly improved manuscript that clearly addresses our (few) concerns. I believe it is suitable for publication in EMBO Reports

2nd Revision - authors' response

19 March 2020

Referee #2:

The authors have provided a detailed response to the comments of the reviewers. However, certain elements have raised concerns and I think that the manuscript needs another round of revision. Overall, this is an interesting study and the manuscript, once revised, should merit publication in EMBO Reports.

Specific points:

1. Fig. 1A shows that a small number of telomeres (about 2 per cell) stain positive for Rad54. Similarly, Fig. 2A shows that only a small number of telomeres (about 2 per cell) stain positive for Rad51. Finally, Fig. 3A shows that only a small number of telomeres (about 2 per cell) stain positive for EdU incorporation. Presumably, it is the same telomeres that stain positive for Rad54, Rad51 and EdU incorporation, but the authors have not demonstrated this.

We did try to do these experiments, but something about the compatibility between the click-it chemistry and the antibodies we used did not allow us to do combined IF/Click-it-chemistry reactions reliably. I'm not sure if the click-it reaction ruined the epitope or just interfered with the signal from the antibody, but those experiments were not fruitful. Unfortunately, at this point due to COVID-19 experiments at Boston University School of Medicine are restricted to essential personnel only and we are unable to follow this up now and have no timeline for our return to the lab. I wish I could be more productive on this front, but we are stuck.

One of the other reviewers raised the legitimate concern that the small number of Edu-positive foci per cell makes it difficult to establish that the number of these foci decreases after Rad54 depletion, since small numbers are associated statistically with higher errors of measurement. This concern is valid and, in fact, the authors reported in the original manuscript that the number of BLM foci decreased after Rad54 depletion, but now they repeated the experiment and report that the number of BLM foci does not decrease. The authors attribute the different result to the method used (antibodies to the endogenous protein versus foci of GFP-BLM), but it is also likely that the different result is due to statistics (large variances due to small number of foci measured).

Here, we aren't counting number of foci, but number of cells positive for colocalization events. For EdU, we define this in the figure legend as "Non-S-phase cells were considered positive if they contained at least one EdU foci colocalizing with telomeres. Data were normalized to mock condition for each repeat". Our analysis suggests that approximately 30-40% of non-sphase cells have at least 1 EdU-Telomere colocalization event, which is consistent with previous publications (Cho Cell 2014). Although depletion of RAD54 does not completely eliminate EdU incorporation the decrease we see is certainly significant. Likewise, for BLM, the number of cells with colocalization events was significant with approximately 80% of cells demonstrating at least 1 BLM-telomere colocalization event within a cell and again this quantification is on par with quantifications from previous publications (Cho Cell 2014).

The key result of the manuscript is shown at Fig. 3b, which shows decreased EdU incorporation at telomeres after RAD54 siRNA. However, the decrease is in the range of 30-40% in 3 of the 4 experiments shown. Given the small number of EdU foci, one begins to wonder if more robust data should be collected. The authors demonstrate in Fig. 1F that expression of TRF1-FOK1 makes all telomeres positive for RAD54 foci. Therefore, expression of TRF1-FOK1 increases dramatically the number of RAD54 foci per cell. I suppose that all the telomeres also become EdU positive, BLM positive and RAD51 positive. Therefore, this is a great system to determine whether RAD54 depletion suppresses EdU incorporation. But the authors did not use this system. Why not?

Therefore, I think that the authors should perform this one experiment, which would allow them to determine with much greater confidence if RAD54 is needed for EdU incorporation at telomeres.

I do believe that the TRF1-FOK1 assay is a great system and has been incredibly useful in progressing the field. However, the system has its limitations. For example, the literature strongly supports replication stress as a contributing factor in the ALT mechanism, yet the TRF1-FOK1 system creates a frank double-strand break. This may significantly alter the mechanism the cell relies on to engage homology directed repair and may not entirely recapitulate true ALT. Thus, using it as the gold standard may not be entirely prudent. It also appears that something may have been lost in the interpretation of the data here. The reviewer states that in Figure 1F we show that 'expression of TRF1-FOK1 makes all telomeres positive for RAD54 foci'. However, the data as quantified in Figure 1G demonstrate that there are certainly cells with zero RAD54-Telomere colocalization events following TRF1-FOK1 induction. Likewise, TRF1-FOK1 induction did not lead to every telomere to become RAD51 positive, as proposed by the reviewer (Cho Cell 2014 and our data Figure EV2B-C). In addition, Cho et al. never showed EdU incorporation at telomeres per se, but showed EdU colocalization with TRF1-FOK1. In our hands, we never saw a significant induction of BrdU at telomeric DNA supporting the idea that there may be some limitations in the use of this assay to completely explain true ALT.

2. In Fig. 2F the authors show the presence of Poleta-GFP foci at telomeres. Presumably, these foci indicate synaptic events at telomeres and one would expect about two foci per cell (same number as the RAD54 foci). This is important, because the authors use Poleta-GFP foci as a surrogate marker for synapsis and accordingly place the function of RAD54 downstream of synapsis. However, Fig. 2F shows that the majority of telomeres (or even all telomeres) stain positive for Poleta-GFP. Therefore, Poleta-GFP may not be a marker of synapsis. Can the authors please comment about this?

Again, I think something may have been lost in the interpretation of the data here. In Figure 2F-G we demonstrate that approximately 20-30% of cells demonstrate at least 1 Pol δ -GFP foci colocalized with telomeric DNA, but certainly not the majority of telomeres in the majority of cells. Thus, there are really only a few colocalization events per cell, not every telomere as suggested by the reviewer.

I am very pleased to accept your manuscript for publication in the next available issue of EMBO reports. Thank you for your contribution to our journal.

Corresponding Author Name: Rachel L. Flynn

Manuscript Number: EMBOR-2019-49495V1